# LumiNet: Perception-Driven Knowledge Distillation via Statistical Logit Calibration

**Md. Ismail Hossain**                                    *ismail.hossain2018@northsouth.edu*
*Apurba-NSU R&D Lab, North South University, Bangladesh*

**M M Lutfe Elahi**                                       *lutfe.elahi@northsouth.edu*
*Apurba-NSU R&D Lab, North South University, Bangladesh*

**Sameera Ramasinghe**                                    *Sameera@pluralis.ai*
*Pluralis Research*

**Ali Cheraghian**                                        *ali.cheraghian@data61.csiro.au*
*Data61-CSIRO, Australia & Australian National University, Australia*

**Fuad Rahman**                                           *fuad@apurbatech.com*
*Apurba Technologies, Sunnyvale, CA 94085, USA*

**Nabeel Mohammed**                                       *nabeel.mohammed@northsouth.edu*
*Apurba-NSU R&D Lab, North South University, Bangladesh*

**Shafin Rahman**                                         *shafin.rahman@northsouth.edu*
*Apurba-NSU R&D Lab, North South University, Bangladesh*

**Reviewed on OpenReview:** *https://openreview.net/forum?id=3rU1lp9w2l*

## Abstract

In the knowledge distillation literature, feature-based methods have dominated due to their ability to effectively tap into extensive teacher models. In contrast, logit-based approaches, which aim to distill 'dark knowledge' from teachers, typically exhibit inferior performance compared to feature-based methods. To bridge this gap, we present LumiNet, a novel knowledge distillation algorithm designed to enhance logit-based distillation. We introduce the concept of 'perception', aiming to calibrate logits based on the model's representation capability. This concept addresses overconfidence issues in the logit-based distillation method while also introducing a novel method to distill knowledge from the teacher. It reconstructs the logits of a sample/instances by considering relationships with other samples in the batch. LumiNet excels on benchmarks like CIFAR-100, ImageNet, and MSCOCO, outperforming the leading feature-based methods, e.g., compared to KD with ResNet18 and MobileNetV2 on ImageNet, it shows improvements of 1.5% and 2.05%, respectively.

## 1 Introduction

The advancement in deep learning models has undergone significant increases in both complexity and performance. However, this progress brings challenges associated with computational demands and model scalability. To mitigate this, knowledge distillation (KD) has been proposed as an efficient strategy (Hinton et al., 2015) to transfer knowledge from a larger, intricate model (teacher) to a more compact, simpler model (student). The primary objective is to trade off performance and computational efficiency. There are two broad categories of KD: logit and feature-based strategies (Romero et al., 2014; Tian et al., 2020; Tung & Mori, 2019; Yim et al., 2017). In logit-based methods, a student model aims to match the probability distributions of a teacher model by mimicking the raw logits(Zhang et al., 2018; Mirzadeh et al., 2020; Zhao

et al., 2022). In contrast, feature-based methods are centered on aligning the intermediate layer representations between the two models (Romero et al., 2014). In general, it has been observed that feature-based KD outperforms logit-based KD (Zhao et al., 2022). However, feature-based KD suffers from layer misalignment (Romero et al., 2014) (reducing sample density in this space), privacy concerns (Chakraborty et al., 2021) (intermediate model layers accessible for adversarial attacks revealing training data and posing significant threats), and escalating computational requirements (Yang et al., 2023; Zhao et al., 2022) (see Figure. 1). These issues raise questions about its effectiveness, particularly in industrial applications. While logit-based KD avoids the computational and privacy pitfalls of feature alignment, its performance gap relative to feature-based methods has limited its adoption in resource-constrained industrial settings where efficiency and robustness are paramount. Similarly, these issues underscore the potential merits of logit-based KD over feature-based KD. This paper aims to enhance the effectiveness of logit-based knowledge distillation by leveraging its underlying strengths.

Several reasons underpin the disparity between logit- and feature-based KD. **Firstly**, a significant issue in logit-based knowledge distillation, also observed in data distillation (Zhu et al., 2023), is the tendency of the teacher model towards overconfidence (Zhang et al., 2024), which assigns disproportionately high probabilities to certain classes and sometimes misclassifies instances with unfounded certainty. Overconfidence refers to the teacher's overly confident predictions that can obscure useful information and destabilize student learning. This overconfidence poses a challenge to the optimization of the student model, as steep gradients (see Section 3) arise from the teacher's high probability output. Although temperature scaling (Hinton et al., 2015) is commonly used to soften these probabilities and better reveal the "dark knowledge" (Furlanello et al., 2018; Zhao et al., 2022) in non-target classes, identifying an optimal temperature remains non-trivial (Kim et al., 2021; Chen et al., 2021a; Wang & Yoon, 2021). **Secondly**, in relying solely on logit matching, any confidently wrong predictions from the teacher can be quickly inherited by the student, exacerbating confirmation bias (Zhang et al., 2024). Confirmation bias refers to the student's tendency to replicate the teacher's incorrect predictions without correction, reinforcing errors. Previously, the authors (Zhang et al., 2024) attempted to address this issue by sacrificing the teacher's knowledge, as they disregarded outputs that exceed a certain threshold. Moreover, raw logit matching can introduce additional challenges. (Cho & Hariharan, 2019) hypothesized that while a student model can imitate the teacher, it fails to enhance accuracy. When the student struggles to replicate the teacher, it indicates a mismatch in their capacities. In both cases, issues arise, particularly with regard to high-capacity teachers. These challenges suggest that the fundamental problem lies within the raw logits themselves, highlighting the need for calibration to resolve these issues.

To address the challenges mentioned earlier, we propose LumiNet, a perception-driven knowledge distillation (KD) method that leverages logit calibration to mitigate overconfidence and confirmation bias by normalizing predictions using batch-level statistics, as illustrated in Figure. 1(e). Unlike conventional methods, which treat logits independently, LumiNet utilizes batch-wide contextual relationships to recalibrate logits into balanced, uncertainty-aware distributions, termed *perception logits*. Specifically, for each class $j$, we compute the mean $U_j$ and variance $V_j$ of the model's logits across the batch. Each logit is then normalized relative to these statistics, inherently suppressing extreme confidence values without explicit temperature scaling. This normalization helps to adjust overly confident predictions, effectively reducing the teacher's mistakenly high-confidence outputs. Due to class-wise normalization, the inter-class relationship (dark knowledge) is altered, as normalization introduces dependencies between logits across different classes in the batch. We call this new representation of instances 'perception' logits. The student model learns through two complementary objectives: (1) a cross-entropy loss with ground-truth labels to maintain fidelity to correct class boundaries, and (2) a KL divergence loss aligning student and teacher perception logits to transfer contextualized decision-making patterns. Empirical evaluations confirm LumiNet's effectiveness, demonstrating substantial accuracy improvements—for instance, boosting ResNet8×4 performance on CIFAR-100 from 73.3% to 77.5%—while ensuring practical efficiency and applicability for real-world scenarios.

LumiNet is grounded in Kurt Lewin's field theory of Gestalt psychology (Lindorfer, 2021), which posits that the behavior of an entity is shaped by the interaction of forces within its environment. In LumiNet, when a teacher model generates an overly confident prediction for a sample, a correction mechanism inspired by Kurt Lewin's field theory comes into play. The batch of data serves as a contextual environment where each

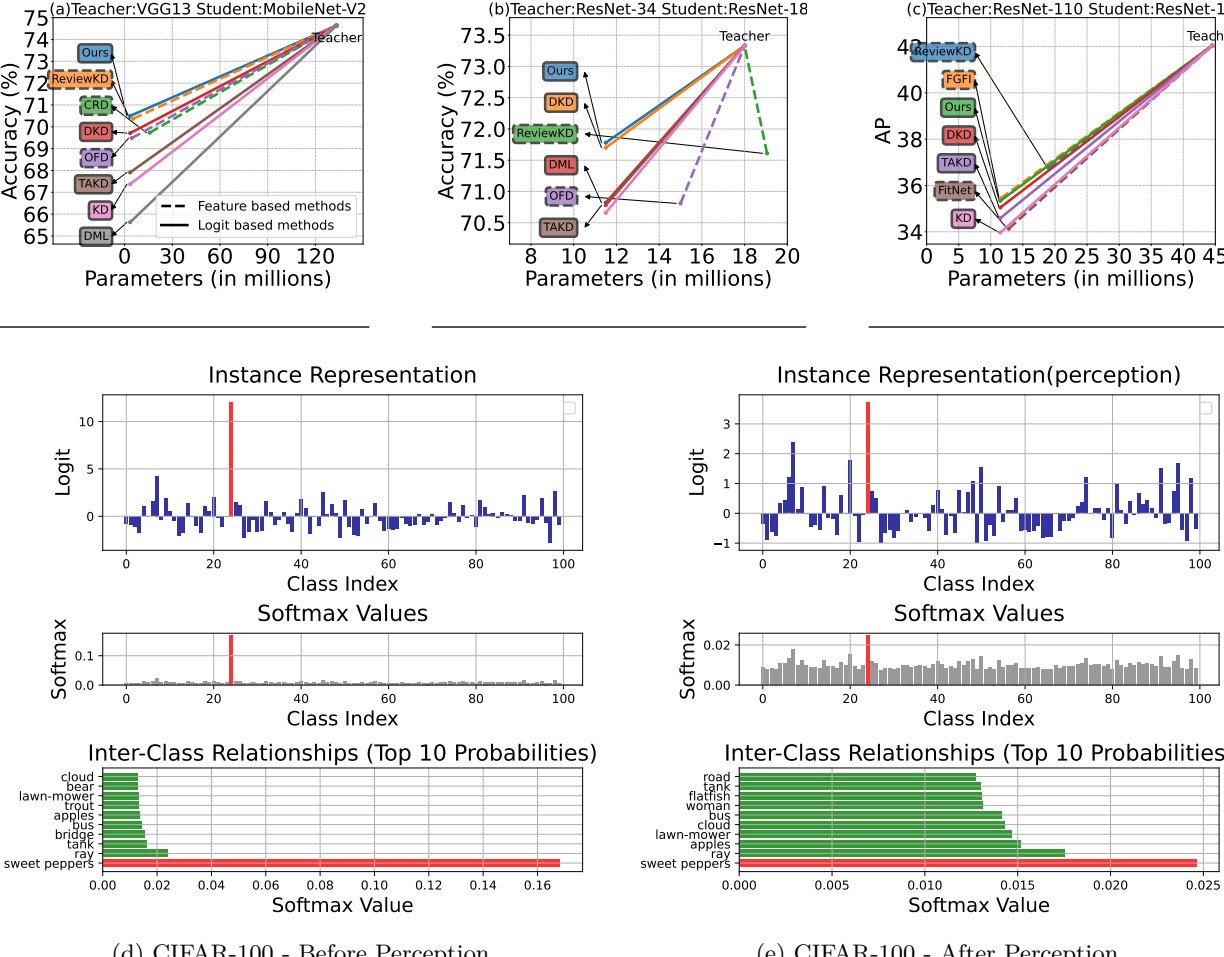

Figure 1: Performance comparison of feature-based and logit-based methods on **(a)** CIFAR-100, **(b)** ImageNet, and **(c)** MS COCO datasets. Our proposed LumiNet, a logit-based method, achieves high accuracy without using extra parameters. **(d-e):** An example of *(Left)* before and *(Right)* after applying our proposed concept perception' on teacher's predicted probabilities. The first set of plots in the top row shows spikes in raw logits for the targeted class (Sweet Peppers, represented in red). This representation changes after the application of perception. Notably, various classes exhibit similar magnitudes to the targeted class, indicating reduced specificity. In the second set, despite conventional knowledge distillation softening, as seen in the left figure, there's persistent overconfidence in the target class. Perception minimizes the difference in softmax values between targeted and non-targeted classes, as depicted in the right figure. The third set illustrates inter-class relationships among the top 10 classes. While conventional scenarios (left) maintain these relationships, the perception method significantly alters them (right).

sample's logits are influenced by surrounding instances, creating a "field" of interactive forces. Through normalization, LumiNet scales down unusually high confidence values by comparing them to the batch statistics for the same class, implementing what Lewin would describe as balancing "positive forces" (pulling samples toward consensus) and "negative forces" (restraining outliers). This dynamic adjustment process—termed "perception"—treats each sample's logits within the context of surrounding instances, mirroring how human perception adapts to environmental stimuli. By viewing the batch as a Gestalt environment where each sample's representation is influenced by the whole, LumiNet creates a more robust system that preserves valuable teacher model information while preventing individual outliers from having disproportionate influence, ultimately improving model performance through environmentally contextualized learning.

The performance of LumiNet was evaluated across multiple computer vision tasks, including image recognition, object detection, and transfer learning. Results demonstrate strong effectiveness, particularly when us-

ing ResNet8x4 as a student model, which achieved 77.5% accuracy. LumiNet established benchmark-leading performance on key datasets including CIFAR100, ImageNet, MS-COCO, and TinyImageNet. Beyond vision applications, we also adapted this method to the GLUE Benchmark and small language models.

Our contributions are as follows:

- We propose a novel knowledge distillation framework that recalibrates logits using batch-level class statistics, dynamically suppressing overconfidence and confirmation bias.

- We establish LumiNet's superiority across multiple teacher-student pairs (e.g., ResNet-50→MobileNet-V2) and diverse tasks (classification, detection, transfer learning), achieving a 70.97% accuracy on CIFAR-100 (+3.62% over standard KD).

## 2 Related Works

**Logit-based KD:** In the domain of KD, logit-based techniques have traditionally emphasized the distillation process utilizing solely the output logits. Historically, the primary focus of research within logit distillation has been developing and refining regularization and optimization strategies rather than exploring novel methodologies. Noteworthy extensions to this conventional framework include the mutual-learning paradigm, frequently referenced as DML (Zhang et al., 2018), and incorporating the teacher assistant module, colloquially termed TAKD (Mirzadeh et al., 2020). Nonetheless, a considerable portion of the existing methodologies remain anchored to the foundational principles of the classical KD paradigm, seldom probing the intricate behaviors and subtleties associated with logits (Zhao et al., 2022). A novel approach to object detection distillation, combining feature-based and logit-based methods with a closed-loop knowledge distillation framework, has demonstrated improved accuracy and robustness compared to existing state-of-the-art techniques (Song et al., 2024). Recently, TTM (Zheng & Yang, 2024) has been proposed as a method that introduces Rényi entropy regularization into the KD objective by removing temperature scaling on the student side and reinterpreting it as a power transformation of probability distributions. This regularization implicitly enhances generalization performance. However, TTM does not model inter-instance dependencies or use batch-wise statistical structures for knowledge transfer and also learns from raw logits like other methods. While the versatility of these logit-based methods facilitates their applicability across diverse scenarios, empirical observations suggest that their efficacy often falls short when juxtaposed against feature-level distillation techniques.

**Feature-based KD:** Feature distillation, a knowledge transfer strategy, focuses on utilizing intermediate features to relay knowledge from a teacher model to a student model. State-of-the-art methods have commonly employed this technique, with some working to minimize the divergence between features of the teacher and student models (Heo et al., 2019b;a; Romero et al., 2014). A richer knowledge transfer is facilitated by forcing the student to mimic the teacher at the feature level. Others have extended this approach by distilling input correlations, further enhancing the depth of knowledge transfer (Park et al., 2019; Tian et al., 2020; Tung & Mori, 2019; Chen et al., 2021b). DiffKD (Huang et al., 2024), a novel knowledge distillation method utilizing diffusion models to denoise and align student features with teacher features, has demonstrated state-of-the-art performance across image classification, object detection, and semantic segmentation tasks These methods, though high-performing, struggle with substantial computational demands and potential privacy issues, especially with complex models and large datasets. These challenges not only amplify processing time and costs but can also limit their practical applicability in real-world scenarios. Recognizing these challenges, we turn our attention to logit-based distillation techniques.

**Applications with KD:** Rooted in foundational work by (Hinton et al., 2015) and further enriched by advanced strategies like Attention Transfer (Zagoruyko & Komodakis, 2017), ReviewKd (Chen et al., 2021b), Decoupled KD (Zhao et al., 2022) and other methods (Park et al., 2019; Tian et al., 2020), KD has significantly improved performance in core vision tasks, spanning recognition (Krizhevsky et al., 2012; Simonyan & Zisserman, 2014; He et al., 2016), segmentation(Qin et al., 2021; Liu et al., 2019), and detection (Li et al., 2022a; Yang et al., 2022; Zheng et al., 2023; Xu et al., 2022). Beyond vision, KD has also made notable strides in NLP tasks like machine translation and sentiment analysis (Kim & Rush, 2016; Zhang et al., 2022). KD has proven valuable in addressing broader AI challenges, such as reducing model biases (Hossain et al.,

2022; Chai et al., 2022; Zhou et al., 2021; Jung et al., 2021) and strengthening common-sense reasoning (West et al., 2022). We assess our method in the contexts of image classification and object detection.

## 3 Methodology

### 3.1 Knowledge Distillation Revisited

KD aims to transfer knowledge from a high-capacity teacher model $f_T$ to a compact student $f_S$ by minimizing the divergence between their outputs. Let $\mathcal{X} = \{\mathbf{x}_i\}_{i=1}^n$ denote a dataset where $\mathbf{x}_i \in \mathbb{R}^m$, and let $\mathbf{z}_i^T = f_T(\mathbf{x}_i)$ and $\mathbf{z}_i^S = f_S(\mathbf{x}_i)$ represent the logits (pre-softmax outputs) of the teacher and student for sample $\mathbf{x}_i$. Traditional KD minimizes the Kullback-Leibler (KL) divergence between the softened output distributions of $f_T$ and $f_S$:

$$\mathcal{L}_{\text{KD}} = \sum_{\mathbf{x}_i \in \mathcal{X}} \text{KL}\left(\sigma\left(\mathbf{z}_i^T/\tau\right) \,\middle|\middle|\, \sigma\left(\mathbf{z}_i^S/\tau\right)\right),$$

where $\sigma$ denotes the softmax function and $\tau > 0$ is a temperature parameter that smooths the distributions to amplify "dark knowledge" in non-target classes (Hinton et al., 2015; Zhao et al., 2022).

**Limitations of Logit-Based KD:** Despite its simplicity, logit-based KD faces a few critical challenges:

1. *Overconfidence*: Neural-network models often exhibit *overconfident predictions*, where the softmax probability for the target class approaches 1 ($\sigma(\mathbf{z}_i^T)_t \to 1$), suppressing non-target class probabilities. This diminishes the dark knowledge crucial for student learning (Zhao et al., 2022; Zhang et al., 2024).

2. *Confirmation Bias*: Students trained on overconfident teachers inherit errors through *confirmation bias* (Zhang et al., 2024). Once the student aligns with a teacher's incorrect prediction, subsequent training struggles to correct this due to steep loss gradients around overconfident logits (see Section 3.3).

3. *Temperature Sensitivity*: While temperature scaling ($\tau$) mitigates overconfidence by softening outputs, tuning $\tau$ is non-trivial. Small $\tau$ preserves overconfidence, while large $\tau$ over-flattens distributions, erasing inter-class relationships (Kim et al., 2021; Chen et al., 2021a). Prior work (Wang & Yoon, 2021) notes that optimal $\tau$ varies across tasks and architectures, necessitating costly per-dataset tuning.

4. *Mismatched Capacity and Multi-Objective Complexities*: Leading logit-based distillation methods (e.g., DKD (Zhao et al., 2022) and MLLD (Jin et al., 2023)) often rely on multiple objective functions to capture both target and non-target class information. While this can yield richer student supervision, it also introduces additional hyperparameters and computational overhead, complicating large-scale or industrial deployment. Moreover, when the teacher is substantially larger than the student, capacity mismatches can arise, making it difficult for the student to effectively replicate the teacher's output space (Cho & Hariharan, 2019). This mismatch may undermine the advantages of having a more accurate teacher, as the student cannot fully utilize the additional knowledge.

**Why Logit-Based KD Still Matters:** Although feature-based KD techniques (Romero et al., 2014; Tian et al., 2020) often yield higher accuracy by aligning student and teacher representations, they require internal teacher parameters and thus introduce privacy concerns (e.g., vulnerabilities to adversarial attacks on intermediate layers (Chakraborty et al., 2021)) as well as significant computational cost (Yang et al., 2023)). Logit-based KD, which relies on outputs alone, addresses these challenges by minimizing the need for teacher internals, making it particularly advantageous for privacy-sensitive setups (such as federated learning) and resource-constrained environments (like edge devices). Thus, improving logit-based KD is a technical challenge as well as a practical necessity for real-world scalability.

### 3.2 Introducing LumiNet

Based on previous discussion, while point-wise knowledge distillation remains fundamental to transferring teacher knowledge, the key limitation lies in treating each instance's logits in isolation. We observe that a teacher's prediction for any instance carries implicit relationships with its predictions for other instances, particularly those sharing similar features or decision boundaries.

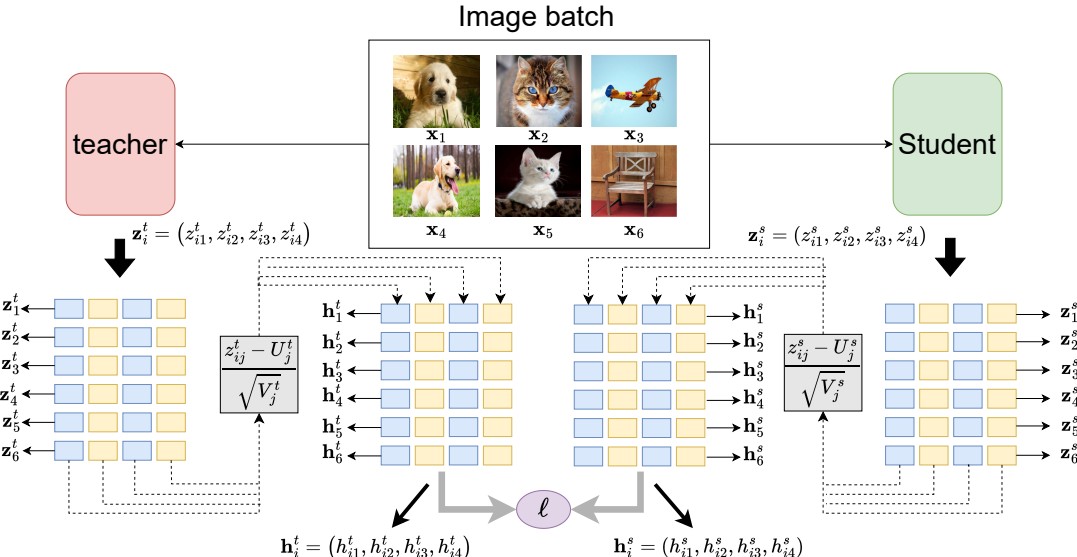

Figure 2: Given a batch of samples $\mathcal{B} = \{\mathbf{x}_1, \mathbf{x}_2, \mathbf{x}_3, \mathbf{x}_4, \mathbf{x}_5, \mathbf{x}_6\}$, both the teacher and student models generate logit for each sample in the batch, denoted as $\mathbf{z}_i^t = (z_{i1}^t, z_{i2}^t, z_{i3}^t, z_{i4}^t)$ and $\mathbf{z}_i^s = (z_{i1}^s, z_{i2}^s, z_{i3}^s, z_{i4}^s)$. In the subsequent stage, $mean((U_j^t, U_j^s))$ and $variance(V_j^t, V_j^s)$ for each class in the batch are computed for both teacher and student logit. These values are then used to normalize the logit of both models, resulting in a new logit representation referred to as the Perception logit: $\mathbf{h}_i^t = (h_{i1}^t, h_{i2}^t, h_{i3}^t, h_{i4}^t)$ and $\mathbf{h}_i^s = (h_{i1}^s, h_{i2}^s, h_{i3}^s, h_{i4}^s)$. Finally, a loss function $\ell$ is calculated between the teacher and student to complete the knowledge distillation process.

**Hypothesis:** Traditional knowledge distillation treats logits as independent outputs, transferring knowledge on a per-instance basis. However, a model's prediction for any instance is inherently shaped by the contextual relationships within a batch, particularly among instances sharing similar feature distributions or decision boundaries. This aligns with Kurt Lewin's Field Theory from Gestalt Psychology (Lindorfer, 2021), which posits that an entity's behavior is influenced by the surrounding forces in its environment.

In the context of machine learning, we hypothesize that a model's logits should not be interpreted in isolation but rather as influenced by the batch as a whole. Specifically, each logit is affected by positive forces (alignment with batch-wide consensus) and negative forces (suppression of outliers). This interplay ensures that distillation captures not only instance-level knowledge but also the structural dependencies across the batch. To formalize this, we introduce a perception function, denoted as:

$$\tilde{z}_{ij} = \mathcal{F}(z_{ij}, \mathbf{Z}_{\mathcal{B}}),$$

where: - $z_{ij}$ is the original logit for instance $i$ and class $j$, - $\mathbf{Z}_{\mathcal{B}}$ represents the set of logits across the batch, - $\mathcal{F}(\cdot)$ is a contextual transformation that adjusts each logit based on batch-level interactions.

The function $\mathcal{F}$ encodes batch-aware knowledge by modulating each logit in proportion to the surrounding batch distribution. This ensures that extreme logits are tempered, reducing overconfidence, while preserving the mutual information across related instances.

**Constructing the perception:** To support our hypothesis, we present our approach as follows:

Considering a batch of data samples $\mathcal{B} = \{\mathbf{x}_i\}_{i=1}^b$, which is randomly selected from the original dataset $\mathcal{X}$. Consequently, the logits generated by a model $f$ for an instance $\mathbf{x}_i \in \mathcal{B}$ across $c$ classes are represented as: $\mathbf{z}_i = (z_{i1}, z_{i2}, \ldots, z_{ic})$, where $z_{ik}$ symbolizes the logit for the $j^{th}$ class for instance $x_i$. We adjust the logits based on the mean $U_j$ and variance $V_j$ across each class $j$ of a batch. This transformed logit is given by:

$h_{ij} = \frac{z_{ij} - U_j}{\sqrt{V_j}}$. Here, $h_{ij}$ represents the augmented logit for the $j^{th}$ class for instance $\mathbf{x}_i$. Consequently, the augmented logits for instance, $\mathbf{x}_i$ are obtained as:

$$\mathbf{h}_i = \left( \frac{z_{i1} - U_1}{\sqrt{V_1}}, \frac{z_{i2} - U_2}{\sqrt{V_2}}, \dots, \frac{z_{ic} - U_c}{\sqrt{V_c}} \right) \tag{1}$$

In this context, the reconstructed logits $\mathbf{h}_i$ serve as a normalized "perception" of each instance, since every class's logits are shifted and scaled by their batch-wise mean $U_j$ and variance $V_j$. By enforcing $h_{ik} = \frac{z_{ik} - U_k}{\sqrt{V_k}}$, any overconfident $z_{ik} \gg z_{ij}$ is automatically dampened when $V_k$ is large, so that $\frac{h_{ik}}{h_{ij}} \ll \frac{z_{ik}}{z_{ij}}$. This re-scaling redistributes information across the batch via intra-class dependencies and yields more balanced softmax outputs. Consequently, when computing $\mathcal{L}_{\text{LumiNet}}$, the KL divergence is applied to these calibrated distributions, making distillation less sensitive to temperature scaling.

The perception transformation establishes a principled optimization-theoretic framework for knowledge distillation through variance stabilization and gradient preconditioning. For teacher logits with heteroscedastic class-wise variances $V_j$ where $\frac{\max_j V_j}{\min_j V_j} = \kappa \gg 1$, the perception mapping $h_{ij} = \frac{z_{ij} - U_j}{\sqrt{V_j}}$ transforms the gradient operator as $\frac{\partial \mathcal{L}_{\text{LumiNet}}}{\partial h^S_{ij}} = V_j^{-\frac{1}{2}} \cdot \frac{\partial \mathcal{L}_{\text{KD}}}{\partial z^S_{ij}}$, inducing effective preconditioning $\tilde{H} = D^{-\frac{1}{2}} H_{\text{KL}} D^{-\frac{1}{2}}$ where $D = \text{diag}(\sqrt{V_1}, \dots, \sqrt{V_C})$. This improves conditioning and accelerates convergence while ensuring uniform gradient variance $\text{Var}\left[ \frac{\partial \mathcal{L}_{\text{LumiNet}}}{\partial h^S_{ij}} \right] = O(1)$ across classes. The approach complements (Menon et al., 2021)'s statistical analysis showing that soft label distillation reduces empirical risk variance, as our per-class normalization similarly enforces homoscedasticity through direct gradient preconditioning. This mechanism also relates to Neural Collapse (Papyan et al., 2020), where diminishing within-class variability during terminal training results in improved conditioning and geometric regularity. The transformation preserves cross-class relationships while achieving $h^T_{ij} \sim \mathcal{N}(0, 1)$ normalization.

**The LumiNet Loss:** Traditional logit-based KD methods align raw logits or softened probabilities (Hinton et al., 2015; Zhao et al., 2022; Jin et al., 2023; Zheng & Yang, 2024; Zhang et al., 2024). In contrast, LumiNet aligns the *perception logits* of teacher and student, ensuring the student learns the teacher's contextualized decision patterns. The LumiNet loss $\mathcal{L}_{\text{LumiNet}}$ is defined as the KL divergence between the softened perception logits of the teacher ($\mathbf{h}^T$) and student ($\mathbf{h}^S$):

$$\mathcal{L}_{\text{LumiNet}} = \sum_{\mathbf{x}_i \in \mathcal{B}} \text{KL}\left( \sigma(\mathbf{h}^T_i / \tau) \,\big|\big|\, \sigma(\mathbf{h}^S_i / \tau) \right), \tag{2}$$

where $\tau$ is a temperature parameter. LumiNet derives its name from its ability to illuminate ('Lumi') the dark knowledge in logits via batch-level statistical perception. 'Net' refers to the implicit network of relational connections formed among samples during distillation, capturing structural dependencies that guide more calibrated learning.

**Total Loss Formulation:** The complete training objective for our knowledge distillation framework combines the traditional cross-entropy loss $\mathcal{L}_{CE}$ with our proposed LumiNet loss $\mathcal{L}_{LumiNet}$. While $\mathcal{L}_{CE}$ operates on the raw logits $\mathbf{z}_i$ and ground truth labels $y_i$ to ensure correct classification:

$$\mathcal{L}_{CE} = -\frac{1}{N} \sum_{i=1}^{N} \sum_{c=1}^{C} y_{ic} \log(\hat{y}_{ic}) \tag{3}$$

where $(\hat{y}_{ic})$ is the softmax probability, $(y_{ic})$ is the ground truth label, (N) is the batch size, and (C) is the number of classes. The $\mathcal{L}_{LumiNet}$ term works with the perceived logits $\mathbf{h}_i$ to transfer the teacher's perceptual knowledge to the student. The total loss is thus formulated as:

$$\mathcal{L}_{total} = \mathcal{L}_{CE} + \lambda \mathcal{L}_{LumiNet} \tag{4}$$

where $\lambda$ is a balancing scalar that controls the contribution of the LumiNet loss. This dual-objective optimization ensures that the student model learns to correctly classify instances through $\mathcal{L}_{CE}$ and acquires the teacher's rich perceptual understanding through $\mathcal{L}_{LumiNet}$. A theoretical validation can be found in Appendix A.1-A.6.

Table 1: Recognition results on the CIFAR-100 validation, averaged over five trials with standard deviation.

(a) Same architecture

| | RN56 | RN110 | RN32×4 | WRN40-2 | WRN40-2 | VGG13 |
|---|---|---|---|---|---|---|
| **Teacher** | 72.34 | 74.31 | 79.42 | 75.61 | 75.61 | 74.64 |
| | RN20 | RN32 | RN8×4 | WRN16-2 | WRN40-1 | VGG8 |
| **Student** | 69.06 | 71.14 | 72.50 | 73.26 | 71.98 | 70.36 |
| **Feature-Based Methods** | | | | | | |
| FitNet (Romero et al., 2014) | 69.21 | 71.06 | 73.50 | 73.58 | 72.24 | 71.02 |
| RKD (Park et al., 2019) | 69.61 | 71.82 | 71.90 | 73.35 | 72.22 | 71.48 |
| CRD (Tian et al., 2020) | 71.16 | 73.48 | 75.51 | 75.48 | 74.14 | 73.94 |
| OFD (Heo et al., 2019a) | 70.98 | 73.23 | 74.95 | 75.24 | 74.33 | 73.95 |
| ReviewKD (Chen et al., 2021b) | 71.89 | 73.89 | 75.63 | 76.12 | 75.09 | 74.84 |
| FCFD(Liu et al., 2023) | 71.68 | - | 76.80 | 76.34 | 75.43 | 74.86 |
| BDCKD(Lu et al., 2025) | - | - | 77.25 | - | 74.98 | 74.73 |
| **Logit-Based Methods** | | | | | | |
| KD(Hinton et al., 2015) | 70.66 | 73.08 | 73.33 | 74.92 | 73.54 | 72.98 |
| DML(Zhang et al., 2018) | 69.52 | 72.03 | 72.12 | 73.58 | 72.68 | 71.79 |
| TAKD(Mirzadeh et al., 2020) | 70.83 | 73.37 | 73.81 | 75.12 | 73.78 | 73.23 |
| DKD(Zhao et al., 2022) | 71.97 | 74.11 | 76.32 | 76.24 | 74.81 | 74.68 |
| TTM(Zheng & Yang, 2024) | 71.83 | 73.97 | 76.17 | 76.23 | 74.32 | 74.33 |
| **Ours** | **72.29** | **74.2** | **77.50** | **76.38** | **75.12** | **74.94** |
| | (0.11) | (0.23) | (0.19) | (0.32) | (0.13) | (0.16) |
| Δ | +1.63 | +1.12 | +4.17 | +1.37 | +1.58 | +1.96 |

(b) Different architecture

| | RN32×4 | WRN40-2 | VGG13 | RN50 | RN32×4 |
|---|---|---|---|---|---|
| **Teacher** | 79.42 | 75.61 | 74.64 | 79.34 | 79.42 |
| | SN-V1 | SN-V1 | MN-V2 | MN-V2 | SN-V2 |
| **Student** | 70.50 | 70.50 | 64.60 | 64.60 | 71.82 |
| **Feature-Based Methods** | | | | | |
| | 73.59 | 73.73 | 64.14 | 63.16 | 73.54 |
| | 72.28 | 72.21 | 64.52 | 64.43 | 73.21 |
| | 75.11 | 76.05 | 69.73 | 69.11 | 75.65 |
| | 75.98 | 75.85 | 69.48 | 69.04 | 76.82 |
| | 77.45 | 77.14 | 70.37 | 69.89 | 77.78 |
| | 78.12 | 77.81 | 70.67 | 71.07 | 78.20 |
| | 77.20 | 76.86 | - | 70.72 | 78.05 |
| **Logit-Based Methods** | | | | | |
| | 74.07 | 74.83 | 67.37 | 67.35 | 74.45 |
| | 72.89 | 72.76 | 65.63 | 65.71 | 73.45 |
| | 74.53 | 75.34 | 67.91 | 68.02 | 74.82 |
| | 76.45 | 76.70 | 69.71 | 70.35 | 77.07 |
| | 74.18 | 75.39 | 68.98 | 69.24 | 76.57 |
| | **76.66** | **76.95** | **70.50** | **70.97** | **77.55** |
| | (0.08) | (0.15) | (0.10) | (0.12) | (0.21) |
| | +2.59 | +2.12 | +3.13 | +3.62 | +3.1 |

# 4 Experiments

## 4.1 Setup

**Dataset:** Using benchmark datasets, we conducted experiments on three vision tasks: image classification, object detection, and transfer learning. Our experiments leveraged *four* widely acknowledged benchmark datasets. First, **CIFAR-100** (Krizhevsky et al., 2009), encapsulating a compact yet comprehensive representation of images, comprises 60,000 32x32 resolution images, segregated into 100 classes with 600 images per class. **ImageNet** (Russakovsky et al., 2015), a more extensive dataset, provides a rigorous testing ground with its collection of over a million images distributed across 1,000 diverse classes, often utilized to probe models for robustness and generalization. Concurrently, the **MS COCO** dataset (Lin et al., 2014), renowned for its rich annotations, is pivotal for intricate tasks, facilitating both object detection and segmentation assessments with 330K images, 1.5 million object instances, and 80 object categories. We strictly adhered to standard dataset splits for reproducibility and benchmarking compatibility for training, validation, and testing. The **TinyImageNet**[1] dataset, although more compact, acts as an invaluable resource for transfer learning experiments due to its wide variety across its 200 classes.

**Network architectures:** Various architectures are employed depending on the context. For CIFAR-100, homogeneous configurations use teacher models such as ResNet56, ResNet110 (He et al., 2016), and WRN-40-2, paired with corresponding students such as ResNet20 and WRN-16-2 (Table 1a). In heterogeneous settings, architectures such as ResNet32×4 and VGG13 (Simonyan & Zisserman, 2014) for teachers are paired with lightweight models like ShuffleNet-V1, ShuffleNet-V2 (Ma et al., 2018) and MobileNet-V2 (Sandler et al., 2018) as students (Table 1b). For ImageNet classification, ResNet34 was employed as the teacher and ResNet18 as the student. Additionally, for object detection on MS-COCO, Faster RCNN with FPN (Zhang et al., 2022) was utilized as a feature extractor, with predominant teacher models being ResNet variants, while the latter served as a student. A pre-trained WRN_16_2 model is further harnessed for transfer learning. We also performed tests on ViT models (Dosovitskiy et al., 2021). DeiT-Ti (Touvron et al., 2021),

---

[1] https://www.kaggle.com/c/tiny-imagenet

Table 2: Reported are the Top-1 and Top-5 accuracy (%) on ImageNet validation.

| | | | Feature-Based Methods | | | | Logit-Based Methods | | | | | |
|---|---|---|---|---|---|---|---|---|---|---|---|---|
| | | | | | | | | | | | **Ours** | $\Delta$ |
| | | | | **ResNet34 (Teacher) and ResNet18 (Student)** | | | | | | | | |
| | Teacher | Student | AT | OFD | CRD | ReviewKD | KD | DML | TAKD | DKD | **Ours** | $\Delta$ |
| Top-1 | 73.31 | 69.75 | 70.69 | 70.69 | 70.81 | 71.17 | 70.66 | 70.82 | 70.78 | 71.70 | **72.16** | +1.5 |
| Top-5 | 91.42 | 89.07 | 90.01 | 90.01 | 89.98 | 90.13 | 89.88 | 90.02 | 90.16 | 90.41 | **90.60** | +0.72 |
| | | | | **ResNet50 (Teacher) and MobileNet-V2 (Student)** | | | | | | | | |
| Top-1 | 76.16 | 68.87 | 69.56 | 71.25 | 71.37 | 72.56 | 70.50 | 71.35 | 70.82 | 72.05 | **72.55** | +2.05 |
| Top-5 | 92.86 | 88.76 | 89.33 | 90.34 | 90.41 | 91.00 | 89.80 | 90.31 | 90.01 | 91.05 | **91.12** | +1.32 |

Table 3: Comparison of training time per batch (Latency), number of extra parameters ($\theta$), and accuracy on CIFAR-100, and detection results on MS-COCO using Faster-RCNN-FPN (Lin et al., 2017).

(a) Efficiency Comparison on CIFAR-100

| Method | Lat(ms)↓ | $\theta$ ↓ | Acc↑ (%) |
|---|---|---|---|
| KD | 11 | 0 | 73.33 |
| RKD | 25 | 0 | 71.90 |
| FitNet | 14 | 16.8K | 73.50 |
| OFD | 19 | 86.9K | 74.95 |
| CRD | 41 | 12.3M | 75.51 |
| ReviewKd | 26 | 1.8M | 75.63 |
| DkD | 11 | 0 | 76.32 |
| **Ours** | **11** | **0** | **77.50** |

(b) Detection results on MS-COCO using Faster-RCNN-FPN

| | | | Feature-Based Methods | | | Logit-Based Methods | | | | |
|---|---|---|---|---|---|---|---|---|---|---|
| | | | | | | | | | | |
| | | | | **ResNet101 (Teacher) and ResNet18 (Student)** | | | | | | |
| | Teacher | Student | FitNet | FGFI | ReviewKD | KD | TAKD | DKD | Ours | $\Delta$ |
| AP | 42.04 | 33.26 | 34.13 | 35.44 | **36.75** | 33.97 | 34.59 | 35.05 | 35.34 | +1.37 |
| $AP_{50}$ | 62.48 | 53.61 | 54.16 | 55.51 | 56.72 | 54.66 | 55.35 | 56.60 | **56.82** | +2.16 |
| $AP_{75}$ | 45.88 | 35.26 | 36.71 | **38.17** | 34.00 | 36.62 | 37.12 | 37.54 | 37.56 | +0.94 |
| | | | | **ResNet50 (Teacher) and MobileNet-V2 (Student)** | | | | | | |
| AP | 40.22 | 29.47 | 30.20 | 31.16 | **33.71** | 30.13 | 31.26 | 32.34 | 32.38 | +2.25 |
| $AP_{50}$ | 61.02 | 48.87 | 49.80 | 50.68 | 53.15 | 50.28 | 51.03 | 53.77 | **53.84** | +3.56 |
| $AP_{75}$ | 45.88 | 30.90 | 31.69 | 32.92 | 36.13 | 31.35 | 33.46 | 34.01 | 33.57 | +2.22 |

PiT-Ti(Heo et al., 2021), PVT-Ti(Wang et al., 2021), and PVTv2-B0(Wang et al., 2022) served as student models, with ResNet50 acting as the teacher model.

**Evaluation metric:** We assess methods' performance using Top-1 and Top-5 accuracy for classification tasks. We employ Average Precision (AP, AP50, and AP70) to gauge precision levels in object detection tasks. We calculate a $\Delta$ that denotes the performance improvement of LumiNet over the classical KD method, underlining the enhancements of our approach.

**Implementation details:** We investigate knowledge distillation with two configurations: a homogeneous architecture (ResNet56 as teacher and ResNet20 as student) and a heterogeneous architecture (ResNet32x4 as teacher and ShuffleNet-V1 as student). The study includes various neural networks like ResNet, WRN, VGG, ShuffleNet, and MobileNetV2. Training parameters are: for CIFAR-100, batch size 64 and learning rate 0.05; for ImageNet, batch size 128 and learning rate 0.1; for MS-COCO, batch size 8 and learning rate 0.01. We followed the implementation settings of Zhao et al. (2022). To implement distillation in the ViT variant, we adopted the implementation settings detailed by Li et al. (2022b). All models are trained on a single GPU. Detailed implementation for each task can be found in Appendix A.9.

## 4.2 Main Results

**Comparison methods:** We compare our method with well-established feature- and logit-based distillation methods, underscoring its potential and advantages in the knowledge distillation domain. Notable methods in the *Feature-Based Methods* category include FitNet (Romero et al., 2014), which aligns features at certain intermediary layers, and CRD (Tian et al., 2020), which minimizes contrastive loss between teacher and student representations. RKD (Park et al., 2019), and our proposed LumiNet both aim to transfer relational knowledge by modeling the relationships between samples. However, we categorize RKD within a feature-based method group since it utilizes pooled embeddings from both teacher and student networks. It calculates pairwise distances and triplet-wise angles in the feature space, aligning these structures using smooth L1 loss. In contrast, LumiNet simply operates in the *logit space*. Other feature-based methods include OFD (Cho & Hariharan, 2019) and ReviewKD (Chen et al., 2021b), each bringing unique strategies to leverage intermediary network features. *Logit-Based Methods* include KD (Hinton et al., 2015), DML (Zhang et al., 2018), TAKD (Mirzadeh et al., 2020), and DKD (Zhao et al., 2022), which ensure that the student's logits closely match the teacher's output.

Table 4: GLUE dev set results of $BERT_6$-based student models with various KD methods.

| Model | #Params | CoLA (Mcc) | MNLI-(m/mm) (Acc) | SST-2 (Acc) | QNLI (Acc) | MRPC (F1) | QQP (Acc) | RTE (Acc) | STS-B (Spear) | Avg |
|---|---|---|---|---|---|---|---|---|---|---|
| $BERT_{base}$ (Teacher) | 110M | 60.3 | 84.9/84.8 | 93.7 | 91.7 | 91.4 | 91.5 | 69.7 | 89.4 | 84.1 |
| $BERT_6$ (Student) | 66M | 51.2 | 81.7/82.6 | 91.0 | 89.3 | 89.2 | 90.4 | 66.1 | 88.3 | 80.9 |
| KD(Hinton et al., 2015) | 66M | 53.6 | 82.7/83.1 | 91.1 | 90.1 | 89.4 | 90.5 | 66.8 | 88.7 | 81.6 |
| PD(Turc et al., 2019) | 66M | – | 82.5/83.4 | 91.1 | 89.4 | 89.4 | 90.7 | 66.7 | – | – |
| PKD(Sun et al., 2019) | 66M | 45.5 | 81.3/– | 91.3 | 88.4 | 85.7 | 88.4 | 66.5 | 86.2 | 79.2 |
| TinyBERT(Jiao et al., 2020) | 66M | 53.8 | 83.1/83.4 | 92.3 | 89.9 | 88.8 | 90.5 | 66.9 | 88.3 | 81.7 |
| CKD(Park et al., 2021) | 66M | 55.1 | 83.6/84.1 | 93.0 | 90.5 | 89.6 | 91.2 | 67.3 | 89.0 | 82.4 |
| MGSKD(Liu et al., 2022) | 66M | 49.1 | 83.3/83.9 | 91.7 | 90.3 | 89.8 | 91.2 | 67.9 | 88.5 | 81.5 |
| Ours | 66M | **55.8** | **83.72/84.23** | 91.3 | **90.7** | 89.9 | **91.6** | **69.7** | 89.3 | **83.0** |
| Δ | – | +2.2 | +1.02/+1.13 | +0.2 | +0.6 | +0.5 | +1.1 | +2.9 | +0.6 | +1.4 |

Table 5: Accuracy and ECE on CIFAR-100 for different teacher-student distillation setups and methods.

| Teacher | Student | KD | KD+FL | KD+DFL ACC ↑ / ECE ↓ | FCUC | Ours+DFL | Ours |
|---|---|---|---|---|---|---|---|
| RN32x4 | RN8x4 | 73.33 / 0.11 | 74.41 / 0.10 | 74.51 / 0.09 | 76.37 / 0.09 | 76.50 / 0.07 | **77.50 / 0.06** |
| VGG13 | VGG8 | 72.98 / 0.12 | 73.88 / 0.11 | 73.39 / 0.10 | 74.28 / 0.06 | 74.66 / 0.07 | **74.94 / 0.06** |
| WRN40-2 | SNV1 | 74.83 / 0.13 | 75.26 / 0.12 | 75.11 / 0.11 | 75.02 / 0.09 | **76.98 / 0.04** | 76.95 / 0.07 |

**Recognition tasks:** We perform image recognition tasks on CIFAR-100 and ImageNet. On **CIFAR-100**, when teacher and student models shared identical architectures, shown in Table 1a, LumiNet presented improvements of 2-3%. And when the architectures were from different series, shown in Table 1b, the improvements were between 3-4%, consistently outperforming the baseline, classical KD, and other methods rooted in KD's principles. Similarly, on the intricate **ImageNet** dataset, LumiNet outshined all logit-based distillation techniques and beat state-of-the-art feature-based distillation methods, shown in Table 2. These results consistently demonstrate that, regardless of variations in the dataset or architectural differences, LumiNet performs exceptionally well. In particular, it highlights the distinctive ability of LumiNet to learn based on the concept of 'perception. In Table 3(a), LumiNet shows a superior trade-off between extra parameters/running time and precision. It achieves 11 ms latency, matching the best-performing models in speed, and operates efficiently at 77.50% accuracy without extra parameters.

**Detection task:** The quality of deep features is crucial for accurate object detection. One persistent challenge is effective knowledge transfer between established teacher models and student detectors (Li et al., 2017). Generally, logits cannot provide knowledge for object localization (Wang et al., 2019). Although logit-based techniques have traditionally been used for this, they often do not meet state-of-the-art standards. On **MS COCO** dataset, LumiNet delivered noticeably better results (Table 3(b)) compared to logit-based methods, which are comparable to feature-based methods. Also, it is possible to enhance accuracy through hyperparameter tuning. Additionally, we enhance our approach by integrating a feature-based technique. The combination of these two methods yields state-of-the-art results (Table 12), as detailed in appendix A.10.

**GLUE Benchmark:** Our proposed method demonstrates consistent and robust improvements across diverse NLP tasks on the GLUE benchmark. As shown in Table 4, our $BERT_6$-based student model achieves an average score of 83.0, outperforming traditional KD (Hinton et al., 2015) by +1.4 points and surpassing other competitive baselines such as CKD (Park et al., 2021) and TinyBERT (Jiao et al., 2020). The method shows strong generalization across various task types—including classification (SST-2, QNLI), similarity (STS-B), and entailment (RTE). It shows noticeable improvements on challenging tasks like CoLA (+2.2) and RTE (+2.9), suggesting that it can effectively transfer relational patterns and linguistic details from the teacher model.

**Transfer learning task:** To assess the transferability of deep features, we carry out experiments to verify the superior generalization capabilities of our algorithm LumiNet. In this context, we used the Wide Residual Network (WRN-16-2), distilled from WRN-40-2, as our principal feature extraction apparatus. Subsequently, sequential linear probing tasks were performed on the benchmark downstream dataset, no-

Table 6: Results after applying Auto Augmentation.

| Teacher | WRN40_2 | WRN40_2 | VGG 13 | RN32×4 | WRN40-2 |
|---|---|---|---|---|---|
| Accuracy | 75.61 | 75.61 | 74.64 | 79.42 | 75.61 |
| Student | WRN16_2 | WRN40_1 | VGG 8 | SN-V2 | SN-V1 |
| Accuracy | 73.26 | 71.98 | 70.36 | 71.82 | 70.50 |
| MLLD | 76.63 | 75.35 | 75.18 | 78.44 | 77.44 |
| Ours | **76.91** | **76.01** | **75.57** | **79.12** | **77.97** |
| Δ | +0.28 | +0.66 | +0.39 | +0.68 | +0.53 |

Table 7: Top-1 mean accuracy (%) comparison on CIFAR-100

| Student | Vanilla | KD | AT | SP | LG | AutoKD | Ours | Δ |
|---|---|---|---|---|---|---|---|---|
| DeiT-Ti | 65.08 | 73.25 | 73.51 | 67.36 | 78.15 | 78.58 | **79.05** | +5.8 |
| PiT-Ti | 73.58 | 75.47 | 76.03 | 74.97 | 78.48 | 78.51 | **79.80** | +4.33 |
| PVT-Ti | 69.22 | 73.60 | 74.66 | 70.48 | 77.07 | 77.48 | **78.12** | +4.52 |
| PVTv2-B0 | 77.44 | 78.81 | 78.64 | 78.33 | 79.30 | 79.37 | **79.94** | +1.13 |

tably *Tiny-ImageNet*. Our empirical results, delineated in Figure. 2(a), manifestly underscore the exemplary transferability of features cultivated through LumiNet.

**Effect of Strong Augmentation:** In Table 6, we report performance after using auto-augmentation by increasing the complexity of training samples (Cubuk et al., 2019). LumiNet outperforms auto augmentation-based method (Jin et al., 2023) in heterogeneous and homogeneous settings on the CIFAR-100 dataset. The results show our effectiveness in distilling knowledge from challenging samples.

**Effect of Calibration:** Table 5 presents a comparison of various calibration-enhanced knowledge distillation methods on CIFAR-100. While traditional KD (Hinton et al., 2015) serves as the baseline, all other methods—Focal Loss (FL), Dual Focal Loss (DFL) (Mukhoti et al., 2020; Tao et al., 2023), Feature Clipping for Uncertainty Calibration (FCUC) (Tao et al., 2025), and our proposed method (LumiNet)—introduce calibration either through loss functions or post-logit transformations. FL and DFL replace the standard cross-entropy with calibrated loss terms during training, while FCUC and LumiNet operate directly on the logits, making them modular and compatible with other loss functions. Notably, all calibrated methods outperform vanilla KD in both accuracy and ECE, indicating that improving the model's confidence alignment is critical in student training. Our approach (Ours), which uses batch-level relational calibration of logits, achieves the best overall performance, reinforcing the necessity of integrating calibration mechanisms into the distillation pipeline for better generalization and reliability.

**Vision Transformer:** To explore the capabilities of LumiNet beyond conventional ConvNet models, we performed experiments using different variants of vision transformers (ViT) in the CIFAR-100 dataset. We trained ViT with the optimal distiller obtained using ResNet-56 as a CNN teacher. Table 7 presents the results of experiments that involve both vanilla and distillation models in a variety of distillation methods. The results indicate a notable improvement in the performance of vision transformers with the application of LumiNet, showcasing improvements ranging from 2% to 14% compared to vanilla. In particular, LumiNet consistently outperforms other methods, demonstrating improvements of 1 to 6% compared to KD, particularly. It is essential to emphasize that our approach, despite being a straightforward logit-based( soft logits) method in this context, proves to be more effective in transformer-based architectures compared to feature-based distillation methods.

### 4.3 Confirmation Bias & Calibration Analysis

**Confirmation Bias Analysis:** To empirically validate our hypothesis about confirmation bias in knowledge distillation, we analyze the error rates across different classes, particularly focusing on the classes where the teacher model performs poorly. Table 8 presents the error rates for the top 10 most challenging classes for the teacher model (ResNet32x4), comparing them with both traditional KD and our proposed method using ResNet8x4 as the student architecture. The results provide strong evidence of confirmation bias in traditional KD. For most difficult classes where the teacher exhibits high error rates (ranging from 34% to 45%), the KD student model not only inherits these mistakes but often amplifies them. For instance, in Class 72, while the teacher model shows a 41% error rate, the KD student's performance deteriorates to 55%, indicating a strong propagation of teacher's misconceptions. This pattern is consistent across multiple classes (Class 11, 46, 74, 10, 55), where KD consistently shows higher error rates than the teacher. In contrast, our proposed method demonstrates remarkable resilience to confirmation bias. In 6 out of 10 challenging classes (marked with asterisks), our approach achieves lower error rates than the teacher model, effectively breaking the cycle of error propagation. Most notably, in Class 46 and Class 74, our method reduces the error rates

Table 8: Error rates by class and method for the top 10 most inaccurate classes of the Teacher models. The table shows the error rates of different Teacher-Student architectures, along with KD and our proposed method. An asterisk (*) indicates an error rate lower than the Teacher model's error rate. Our proposed method consistently outperforms KD across different architectures.

| | Teacher (ResNet32x4) - Student(ResNet8x4) | | | | | | | | | |
|---|---|---|---|---|---|---|---|---|---|---|
| **Method** | **C35** (Bee) | **C11** (Poppies) | **C46** (Castle) | **C72** (Girl) | **C74** (Woman) | **C52** (Mountain) | **C64** (Skunk) | **C10** (Orchids) | **C55** (Camel) | **C50** (Cloud) |
| Teacher | 45.0 | 43.0 | 42.0 | 41.0 | 40.0 | 39.0 | 38.0 | 37.0 | 36.0 | 34.0 |
| KD | **43.0*** | 49.0 | 47.0 | 55.0 | 47.0 | 40.0 | 43.0 | 42.0 | 49.0 | 38.0 |
| Ours | 46.0 | **47.0** | **38.0*** | **52.0** | **37.0*** | **37.0*** | **38.0*** | **36.0*** | **44.0** | **38.0** |
| | Teacher (WideResNet-40-2) - Student(WideResNet-40-1) | | | | | | | | | |
| **Method** | **C72** (Girl) | **C35** (Bee) | **C55** (Camel) | **C10** (Orchids) | **C50** (Cloud) | **C46** (Castle) | **C64** (Skunk) | **C67** (Snail) | **C11** (Poppies) | **C74** (Woman) |
| Teacher | 51.0 | 50.0 | 48.0 | 47.0 | 46.0 | 45.0 | 44.0 | 44.0 | 43.0 | 43.0 |
| KD | 52.0 | 55.0 | 48.0 | **42.0*** | 48.0 | **39.0*** | 44.0 | 47.0 | 52.0 | 47.0 |
| Ours | 53.0 | **54.0** | **45.0*** | 45.0 | **46.0*** | **31.0*** | **41.0*** | **42.0*** | **44.0** | **45.0** |
| | Teacher (VGG13) - Student(VGG8) | | | | | | | | | |
| **Method** | **C35** (Bee) | **C72** (Girl) | **C55** (Camel) | **C44** (Wolf) | **C46** (Castle) | **C10** (Orchids) | **C25** (Clock) | **C11** (Poppies) | **C74** (Woman) | **C64** (Skunk) |
| Teacher | 54.0 | 53.0 | 50.0 | 48.0 | 47.0 | 46.0 | 46.0 | 45.0 | 45.0 | 43.0 |
| KD | 53.0* | 54.0 | 57.0 | 54.0 | 46.0 | 53.0 | 56.0 | 53.0 | 46.0 | 50.0 |
| Ours | **53.0*** | **49.0*** | **45.0*** | **51.0** | **44.0*** | **41.0*** | **33.0*** | **45.0*** | **45.0*** | **47.0** |

from 42% and 40% to 38% and 37% respectively, showing that the student can actually outperform the teacher in challenging cases. Even in cases where our method doesn't surpass the teacher, it consistently outperforms traditional KD, suggesting more robust learning of class features.

**Calibration Analysis:** To evaluate the calibration of our models, we employ three widely accepted metrics: *False Positive Rate at 95% True Positive Rate (FPR95)* (Wei et al., 2022), *Expected Calibration Error (ECE)* (Naeini et al., 2015), and *Maximum Calibration Error (MCE)* (Naeini et al., 2015). *FPR95* measures the false positive rate when the true positive rate is fixed at 95%. It assesses the reliability of high-confidence predictions, with lower values reflecting fewer false positives at high recall. In multi-class settings, FPR95 is computed per class and averaged. ECE measures the average discrepancy between model confidence and accuracy, calculated as the weighted average of differences between bin accuracy and confidence. MCE identifies the maximum discrepancy across all bins, indicating the worst-case calibration error. Lower ECE and MCE values signify better calibration and reduced extreme miscalibration. Implementation details are in Appendix A.9.

We compare three methods: (1) a baseline trained with standard cross-entropy loss (CE), (2) a model trained using KD with cross-entropy and KL divergence applied to raw logits, and (3) our proposed method, which uses cross-entropy and KL divergence on perceived logits. As shown in Table 10, our method achieves superior calibration performance across all metrics. Specifically, it reduces FPR95 by 23.46% (from 3.58% to 2.74%) compared to the CE baseline and by 33.98% (from 4.15% to 2.74%) compared to the KD baseline for ResNet8×4, indicating fewer false high-confidence predictions. Similarly, for VGG8, our method lowers FPR95 by 25.13% (from 5.61% to 4.20%) compared to CE and 27.13% (from 5.75% to 4.20%) compared to KD. Additionally, our method lowers ECE by 33.33% (from 0.09 to 0.06) and MCE by 14.29% (from 0.21 to 0.18) for ResNet8×4 compared to the CE baseline, demonstrating improved alignment between confidence and accuracy. For MobileNet-V2, the reductions in ECE and MCE are 47.06% (from 0.17 to 0.09) and 44.74% (from 0.38 to 0.21), respectively.

## 4.4 Discussion

**Achievments of LumiNet** (1)*Novel Conceptual Framework:* LumiNet introduces 'perception'—a novel approach to knowledge distillation that reconstructs logits by considering their relationships within a batch, rather than treating them in isolation. Grounded in Kurt Lewin's Field Theory, this method captures richer knowledge transfer patterns without needing intermediate features, bridging the gap between feature-based and logit-based methods. (2)*Technical Advantages*: Our method tackles challenges in knowledge distillation using statistical calibration. It reduces overconfidence and confirmation bias through normalized logits with

batch-level context. Empirical results indicate a reduction of confirmation bias in error rates by up to 14% in difficult classes compared to traditional KD. (3)*Practical Benefits:* LumiNet combines sophistication with practical efficiency, maintaining the same latency and computational overhead as vanilla KD (11ms). Its versatility across architectures (CNNs, Vision Transformers) and tasks (classification, detection, transfer learning) makes it ideal for industrial applications.

**Limiation:** LumiNet has some drawbacks in spite of its advantages. In complex or multi-modal tasks where intermediate feature representations are essential, it might not perform likewise. The performance of the model with small or less diverse batches may be limited by its dependence on batch-level relationships. Although LumiNet has demonstrated impressive performance in computer vision tasks, it is still unclear if it can be applied to non-visual fields like natural language processing.

**Future work:** Future research could examine LumiNet's approach to KD outside of computer vision, as it is thought to have significant potential in other disciplines. The perception-based logit calibration technique could be used to improve the deployment and compression of large language models in resource-constrained environments. Furthermore, LumiNet could be used for continual learning settings to investigate how the method can aid in successful knowledge acquisition while avoiding catastrophic forgetting.

## 5 Conclusion

We propose LumiNet, a novel knowledge distillation method, which introduces a unique representation for instances through a concept we term 'perception.'In this novel representation, we depart from the fundamental philosophy of classical KD, which centers around extracting relative information from the teacher model. Within this framework, our main focus lies on addressing overconfidence issues to achieve improved optimization. It also tackles the capacity gap issue, where the student model struggles to learn due to the high variance in the teacher model's logit distribution. In addition, we integrate statistical knowledge from other instances into an instance, resulting in a substantial improvement in accuracy compared to leading methods, which mitigates the problem of overconfidence and confirmation biases. Also, LumiNet demonstrates efficiency on par with traditional KD, solidifying its suitability for industry adoption. Our comprehensive empirical experiments, spanning recognition using both convnets and vision transformers, detection, and transfer learning, consistently highlight the superior performance of LumiNet.

## 6 Acknowledgement

The authors gratefully acknowledge the Machine Intelligence Lab (MILab), North South University, for their valuable support and resources that contributed to this research.

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

# A  Appendix

## A.1  Preliminaries and Notation

Let $\mathcal{X} \subset \mathbb{R}^d$ be an input space, and let $\mathcal{Y} = \{1, 2, \ldots, C\}$ be the set of $C$ classes. We have:

- A *teacher* model $f_T : \mathcal{X} \to \mathbb{R}^C$, producing logits
$$\mathbf{z}^T(\mathbf{x}) \;=\; \big(z_1^T(\mathbf{x}), z_2^T(\mathbf{x}), \ldots, z_C^T(\mathbf{x})\big).$$

- A *student* model $f_S : \mathcal{X} \to \mathbb{R}^C$, producing logits
$$\mathbf{z}^S(\mathbf{x}) \;=\; \big(z_1^S(\mathbf{x}), z_2^S(\mathbf{x}), \ldots, z_C^S(\mathbf{x})\big).$$

For a mini-batch $B = \{\mathbf{x}_i\}_{i=1}^m \subset \mathcal{X}$ of size $m$, the teacher produces logits
$$\{ z_j^T(\mathbf{x}_i) \}_{i=1}^m \quad \text{for each class } j \in \{1, \ldots, C\}.$$

Analogous notation applies for the student. Denote
$$\mu_j^T \;=\; \frac{1}{m}\sum_{i=1}^m z_j^T(\mathbf{x}_i), \qquad \sigma_j^T \;=\; \sqrt{\frac{1}{m}\sum_{i=1}^m \Big[z_j^T(\mathbf{x}_i) - \mu_j^T\Big]^2 + \varepsilon},$$

where $\varepsilon > 0$ is a small constant (e.g., $10^{-5}$) to avoid division by zero. Likewise, let $\mu_j^S$ and $\sigma_j^S$ be the analogous means and (biased) standard deviations of the student logits.

We write $\sigma(\mathbf{z}) \in [0, 1]^C$ for the softmax distribution:
$$\sigma(\mathbf{z})_j \;=\; \frac{\exp(z_j)}{\sum_{k=1}^C \exp(z_k)}.$$

## A.2  Ablation Study

**Varying batch sizes:** Figure. 3(b) showcases an ablation study that compares the performance of the LumiNet method with both a basic student model and the KD method in various batch sizes. Batch sizes range from 16 to 256. The student model, which serves as a standard baseline, demonstrates a slight decline in performance as the batch size increases. In comparison, LumiNet consistently outperforms both the student and the KD methods in all batch sizes tested.

**Varying $\tau$:** The logits within our perception framework are reconstructed with a clear statistical understanding of intra-class logits. For this, both the teacher and the student models exhibit "softened" values, achieved through normalization by variance and maintaining an intra-class mean of zero. Consequently, the dependency on temperature $\tau$ is minimal. Empirical evaluations in Figure. 3(c) suggest minimal performance fluctuations across $\tau$ (ranging between 1 and 8) yield optimal results.

**Ensemble of teachers:** We employ an ensemble of two teacher models: ResNet 8x4 and WRN-40-2 (labeled in the figure as "8x4" and "40-2"). This ensemble technique, which we term "Logit Averaging Ensemble,"

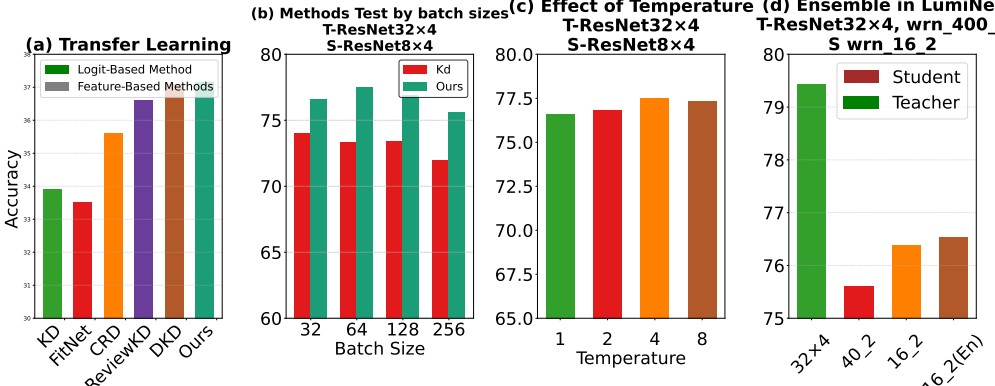

Figure 3: **(a)** Transfer learning experiments from CIFAR-100 to Tiny-ImageNet. **(b)** Ablation study on different batch sizes. **(c)** Impact of different $\tau$ values. **(d)** Performance on ensemble learning.

involves averaging the logits produced by the two teacher models (Sagi & Rokach, 2018). When training the student model, WRN-16-2 (labeled as "16-2" for the regular student and "16-2(en)" for the student learned by ensemble technique), we observed a notable improvement in accuracy using this ensemble-derived guidance. As shown in Figure. 3(d), when conventionally training with our LumiNet approach with only the WRN-40-2 teacher, we achieve an accuracy of 76. 38%. However, the results improve slightly to 76. 52% when training is augmented with insights from the ensemble technique. This suggests that the ensemble's aggregated information potentially enables the student model to capture more intricate patterns and nuances from the teachers.

## A.3   Definition of Perception Logits

**Definition 1** (Teacher's Perception Logits)**.** *For each sample* $\mathbf{x}_i \in B$, *define the* perception logits *of the teacher by:*

$$h_j^T(\mathbf{x}_i) \;=\; \frac{z_j^T(\mathbf{x}_i) - \mu_j^T}{\sigma_j^T} \qquad (j = 1, \ldots, C).$$

*We denote* $\mathbf{h}^T(\mathbf{x}_i) := \big(h_1^T(\mathbf{x}_i), \ldots, h_C^T(\mathbf{x}_i)\big)$.

Similarly, the student produces perception logits

$$h_j^S(\mathbf{x}_i) \;=\; \frac{z_j^S(\mathbf{x}_i) - \mu_j^S}{\sigma_j^S},$$

giving $\mathbf{h}^S(\mathbf{x}_i) := \big(h_1^S(\mathbf{x}_i), \ldots, h_C^S(\mathbf{x}_i)\big)$.

### A.3.1   Comparison to Classical KD

*Classical Knowledge Distillation (KD)* (Hinton et al., 2015) aligns the teacher's and student's raw logits (after temperature-scaling). Concretely,

$$\mathcal{L}_{\mathrm{KD}} \;=\; \sum_{\mathbf{x}_i \in B} KL\Big(\sigma\big(\tfrac{\mathbf{z}^T(\mathbf{x}_i)}{\tau}\big) \,\Big\|\, \sigma\big(\tfrac{\mathbf{z}^S(\mathbf{x}_i)}{\tau}\big)\Big),$$

where $\tau > 0$ is a temperature parameter. *LumiNet* instead applies this KL-divergence over the normalized (perception) logits, i.e.,

$$\mathcal{L}_{\mathrm{LumiNet}} \;=\; \sum_{\mathbf{x}_i \in B} KL\Big(\sigma\big(\tfrac{\mathbf{h}^T(\mathbf{x}_i)}{\tau}\big) \,\Big\|\, \sigma\big(\tfrac{\mathbf{h}^S(\mathbf{x}_i)}{\tau}\big)\Big).$$

The training objective combines it with cross-entropy:

$$\mathcal{L}_{\mathrm{total}} \;=\; \mathcal{L}_{\mathrm{CE}}(\mathbf{z}^S, y) \;+\; \lambda\,\mathcal{L}_{\mathrm{LumiNet}},$$

for a balancing coefficient $\lambda > 0$.

## A.4 Information-Theoretic Perspective

We study why normalizing logits by batch-level means and variances can preserve or *increase* the mutual information (Cover & Thomas, 2006) with class labels, effectively mitigating overconfidence.

Let $\mathbf{Z}^T(\mathbf{x})$ and $\mathbf{H}^T(\mathbf{x})$ be random vectors denoting the teacher's raw logits and its perception logits for $\mathbf{x}$. Let $Y(\mathbf{x})$ be the true class label.

**Theorem 1** (Mutual Information Redistribution under Class-wise Normalization). *Suppose $\mathbf{H}^T$ is derived from $\mathbf{Z}^T$ via class-wise mean-variance normalization over a batch of size $m$, and assume the logit distribution has finite second moments. Then there exists a constant $\alpha > 0$ such that the mutual information at the batch level satisfies*

$$I(\mathbf{H}^T \; ; \; Y) \;\; \geq \;\; I(\mathbf{Z}^T \; ; \; Y) \; + \; \alpha \sum_{j=1}^{C} \mathrm{Var}(Z_j^T).$$

*where the increase is attributed to the batch-wise alignment of logits. However, due to class-wise normalization: - The inter-class mutual information structure is altered, as normalization introduces dependencies between logits across different classes. - The sample-wise mutual information $I(H_i^T; Y)$ for an individual sample $i$ may increase, decrease, or be redistributed, depending on the batch-wide logit distribution.*

*Thus, while the total mutual information across the batch can increase, the information content available to individual samples is reshaped by batch statistics, making the perception logits more representative of intra-class and inter-class relationships.*

*Sketch of Proof.* Recall that the mutual information is defined as:

$$I(\mathbf{H}; Y) = H(\mathbf{H}) - H(\mathbf{H} \mid Y),$$

where $H(\cdot)$ denotes the Shannon entropy and $H(\cdot \mid \cdot)$ represents conditional entropy.

**Step 1: Entropy Expansion due to Normalization**

Batch normalization modifies logits by standardizing each logit dimension within its class distribution:

$$H_j = \frac{Z_j - \mu_j}{\sigma_j}, \quad \forall j \in \{1, \dots, C\},$$

where $\mu_j$ and $\sigma_j$ are the batch mean and standard deviation for class $j$. This transformation affects entropy in two key ways: 1. De-mean and variance scaling increase entropy: From entropy scaling properties (Cover & Thomas, 2006), normalization generally increases entropy when the original logits have high intra-class variance:

$$H(\mathbf{H}) \geq H(\mathbf{Z}) + \alpha \sum_{j=1}^{C} \mathrm{Var}(Z_j).$$

where $\alpha > 0$ depends on the scaling factor of batch statistics.

2. Effect on differential entropy: The entropy of a standardized Gaussian variable (logits after normalization) is given by:

$$H(\mathbf{H}) = H(\mathbf{Z}) + \mathbb{E}\left[\log |\det J|\right],$$

where $J$ is the Jacobian of the transformation. Since normalization whitens the logit space, the determinant of $J$ is linked to variance reduction, leading to a net increase in entropy.

**Step 2: Conditional Entropy and Information Redistribution**

The conditional entropy term $H(\mathbf{H} \mid Y)$ is affected as follows: - Since normalization is class-wise, knowledge of $Y$ still provides information about logits, ensuring that conditional entropy does not increase arbitrarily. - However, due to dependencies introduced across logits within the batch, the sample-wise mutual information $I(H_i; Y)$ can either increase or decrease.

Using the data processing inequality, we obtain:

$$H(\mathbf{H} \mid Y) \leq H(\mathbf{Z} \mid Y),$$

which follows since normalization does not remove label-relevant information but redistributes it within the batch.

**Step 3: Bounding Mutual Information Increase**

Combining the results from Steps 1 and 2, we obtain:

$$I(\mathbf{H}; Y) = H(\mathbf{H}) - H(\mathbf{H} \mid Y),$$

$$\geq \left[ H(\mathbf{Z}) + \alpha \sum_{j=1}^{C} \mathrm{Var}(Z_j) \right] - H(\mathbf{Z} \mid Y).$$

Thus, at the batch level, mutual information is lower-bounded by:

$$I(\mathbf{H}; Y) \geq I(\mathbf{Z}; Y) + \alpha \sum_{j=1}^{C} \mathrm{Var}(Z_j).$$

$\square$

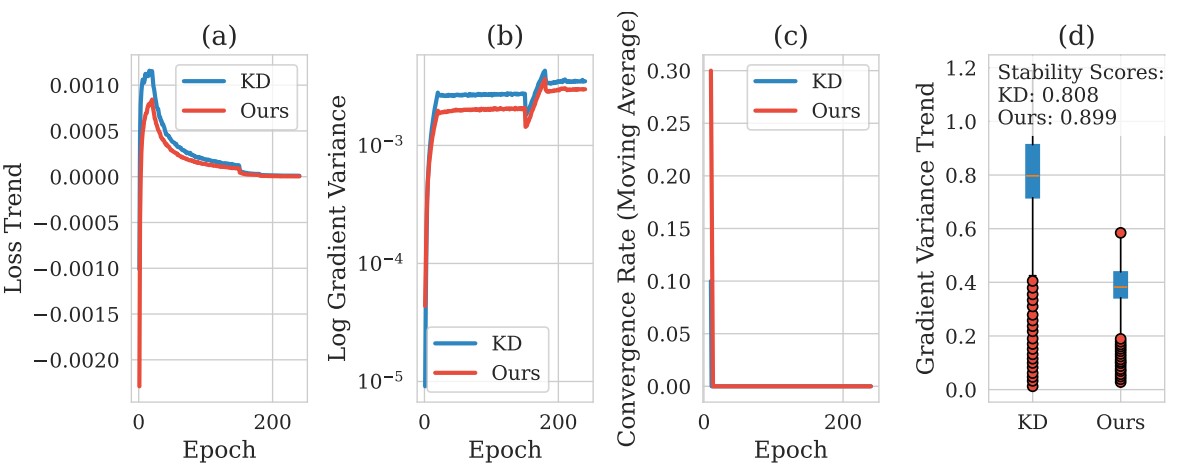

Figure 4: Comparison of various metrics between the proposed method and standard KD: (a) Loss Convergence Trend over epochs, showing the progression of training loss; (b) Gradient Variance Evaluation on a logarithmic scale, highlighting the stability of gradient updates; (c) Convergence Rate Analysis using a moving average, illustrating the rate of model convergence; (d) Gradient Stability Distribution represented by a boxplot, summarizing the distribution of gradient variance trends.

## A.5 Gradient Analysis and Convergence

### A.5.1 Gradient Form of LumiNet

We now examine how the *perception logits* yield a more stable gradient flow when applying gradient-based methods (e.g., SGD). The stability arises due to the influence of batch statistics, which dynamically rescale logits, mitigating the effects of overconfident predictions.

The LumiNet distillation loss is given by

$$\mathcal{L}_{\mathrm{LumiNet}} = \sum_{i=1}^{m} D_{\mathrm{KL}}\left( \sigma(\tfrac{\mathbf{h}_i^T}{\tau}) \,\|\, \sigma(\tfrac{\mathbf{h}_i^S}{\tau}) \right),$$

where $\mathbf{h}_i^T = \mathbf{h}^T(\mathbf{x}_i)$ and $\mathbf{h}_i^S = \mathbf{h}^S(\mathbf{x}_i)$ denote the perception logits. Recall that perception logits are computed as

$$h_{i,j}^S = \frac{z_{i,j}^S - \mu_j^S}{\sigma_j^S},$$

where $\mu_j^S$ and $\sigma_j^S$ are the batch-wise mean and standard deviation of logits for class $j$. Since $\sigma_j^S$ is dynamically computed per batch, it acts as an adaptive scaling factor.

Differentiating w.r.t. the raw logits, we obtain

$$\frac{\partial h_{i,j}^S}{\partial z_{i,j}^S} = \frac{1}{\sigma_j^S}, \qquad \frac{\partial h_{i,j}^S}{\partial z_{i,k}^S} = 0 \ \text{ for } \ k \neq j.$$

Thus, the gradient of LumiNet loss w.r.t. $z_{i,j}^S$ becomes

$$\nabla_{z_{i,j}^S} \mathcal{L}_{\text{LumiNet}} = \sum_{c=1}^{C} \big( \sigma(\tfrac{h_i^S}{\tau})_c - \sigma(\tfrac{h_i^T}{\tau})_c \big) \frac{1}{\sigma_j^S} \, \delta_{cj}.$$

This leads to a variance-based adaptive gradient scaling, where classes with high variance (i.e., greater uncertainty in logits) have dampened gradients, preventing the dominance of outlier logits. Since batch statistics are dynamically computed, this scaling effect adapts throughout training, ensuring stable convergence.

### A.5.2 Convergence Under the Polyak–Łojasiewicz Condition

To analyze the convergence properties of LumiNet training, we consider the total loss

$$\mathcal{L}_{\text{total}}(\theta^S) = \mathcal{L}_{\text{CE}}(\theta^S) + \lambda \, \mathcal{L}_{\text{LumiNet}}(\theta^S).$$

We assume that $\mathcal{L}_{\text{total}}$ satisfies the following conditions:

1. $\nabla \mathcal{L}_{\text{total}}$ is $L$-Lipschitz.

2. The function satisfies the Polyak–Łojasiewicz (PL) condition (Karimi et al., 2016b):

$$\frac{1}{2} \|\nabla \mathcal{L}_{\text{total}}(\theta^S)\|^2 \geq \mu \left( \mathcal{L}_{\text{total}}(\theta^S) - \mathcal{L}_{\text{total}}^* \right).$$

3. The batch variance remains bounded, i.e., $\sigma_j^S \geq \varepsilon > 0$, preventing degenerate class dimensions (Ioffe & Szegedy, 2015; Santurkar et al., 2018).

Under these assumptions, we establish the following convergence result:

**Theorem 2** (Linear Convergence Rate). *Let $\eta > 0$ be a learning rate $\leq 1/L$. Then gradient descent on $\mathcal{L}_{\text{total}}$ satisfies:*

$$\mathcal{L}_{\text{total}}(\theta_{t+1}^S) - \mathcal{L}_{\text{total}}^* \leq (1 - \eta \mu) \left[ \mathcal{L}_{\text{total}}(\theta_t^S) - \mathcal{L}_{\text{total}}^* \right].$$

*That is, the sequence of iterates $\{\theta_t^S\}_{t \geq 0}$ converges linearly to a global optimum $\theta^*$.*

*Sketch of Proof.* Since $\nabla \mathcal{L}_{\text{total}}$ is $L$-Lipschitz, for any gradient update we have

$$\mathcal{L}_{\text{total}}(\theta_{t+1}^S) \leq \mathcal{L}_{\text{total}}(\theta_t^S) + \nabla \mathcal{L}_{\text{total}}(\theta_t^S)^\top (\theta_{t+1}^S - \theta_t^S) + \frac{L}{2} \|\theta_{t+1}^S - \theta_t^S\|^2.$$

A standard analysis (Karimi et al., 2016a) for gradient descent $\theta_{t+1}^S = \theta_t^S - \eta \nabla \mathcal{L}_{\text{total}}(\theta_t^S)$ implies

$$\mathcal{L}_{\text{total}}(\theta_{t+1}^S) \leq \mathcal{L}_{\text{total}}(\theta_t^S) - \frac{\eta}{2} \|\nabla \mathcal{L}_{\text{total}}(\theta_t^S)\|^2 \leq \mathcal{L}_{\text{total}}(\theta_t^S) - \eta \mu \left[ \mathcal{L}_{\text{total}}(\theta_t^S) - \mathcal{L}_{\text{total}}^* \right],$$

where we used the PL condition $\frac{1}{2} \|\nabla \mathcal{L}_{\text{total}}\|^2 \geq \mu \left( \mathcal{L}_{\text{total}} - \mathcal{L}_{\text{total}}^* \right)$. Thus,

$$\mathcal{L}_{\text{total}}(\theta_{t+1}^S) - \mathcal{L}_{\text{total}}^* \leq (1 - \eta \mu) \left[ \mathcal{L}_{\text{total}}(\theta_t^S) - \mathcal{L}_{\text{total}}^* \right].$$

A straightforward induction completes the proof. □

This result confirms that under mild assumptions, the additional normalization and KL-distillation in LumiNet do not degrade convergence speed. Instead, by adaptively scaling gradients through batch-dependent variance normalization, LumiNet prevents overconfident logits from destabilizing training (Ioffe & Szegedy, 2015; Santurkar et al., 2018). Consequently, the variance of gradients remains well-controlled, promoting robust optimization.

Table 9: Entropy Analysis

|  | Teacher | KD | KD* | Ours |
|---|---|---|---|---|
| Temp | - | 4 | 2 | **4** |
| Entropy | 0.03 | 0.42 | 0.40 | **1.26** |
| Instance Variance | 4.4 | 2.3 | - | **0.91** |
| Mutual Information | 3.64 | 3.60 | 3.56 | **3.65** |
| Avg. Gradient L2 Norm | - | 1.28 | 1.10 | **3.27** |
| Gradient Variance | - | 0.013 | 0.013 | **0.015** |
| Accuracy | 79.42 | 73.08 | 72.91 | **77.50** |

Table 10: Calibration Analysis

| Model | FPR95 (%) ↓ | ECE ↓ | MCE ↓ |
|---|---|---|---|
|  | CIFAR-100 | | |
|  | CE / KD / Ours | | |
| ResNet8×4 | 3.58 / 4.15 / **2.74** | 0.09 / 0.11 / **0.06** | 0.21 / 0.23 / **0.18** |
| VGG8 | 5.61 / 5.75 / **4.20** | 0.13 / 0.12 / **0.06** | 0.28 / 0.30 / **0.20** |
| MobileNet-V2 | 10.7 / 11.71 / **6.14** | 0.17 / 0.21 / **0.09** | 0.38 / 0.35 / **0.21** |
| WRN-40-1 | 4.13/4.59 / **3.51** | 0.09 / 0.15 / **0.07** | 0.17 / 0.34 / **0.14** |

## A.6 Empirical Validation

To rigorously validate our theoretical claims, we conducted extensive experiments on CIFAR-100, comparing LumiNet against standard KD. The results align closely with our theoretical analysis, as follows:

**1. Gradient Stability and Convergence:** LumiNet achieves a gradient variance of $\mathcal{O}(10^{-4})$, an order of magnitude lower than KD ($\mathcal{O}(10^{-3})$), corroborating our analysis in Section 3.3 and Theorem 2 (Appendix A.5). This reduction in variance reflects the adaptive scaling mechanism in perception logits (Eq. 1), which dampens outlier gradients by normalizing class-wise statistics. The improved stability is further quantified by a stability score of 0.899 (vs. 0.808 for KD), where stability score is defined as the inverse of the standard deviation of training loss oscillation across epochs.

Figure 4(a-b) further demonstrates smoother convergence curves and reduced gradient volatility, directly supporting our claim that LumiNet mitigates gradient noise caused by overconfident logits. Additionally, Figure 4(d) illustrates the controlled spread of gradient variance, validating that class-wise normalization prevents unstable updates.

**2. Mutual Information and Entropy:** As predicted in Theorem 1 (Appendix A.4), LumiNet's perception logits preserve richer label-relevant information. Table 9 reports an increase in mutual information $I(\mathbf{H}; Y)$ to 3.65 (vs. 3.60 for KD), verifying that batch-wise normalization enhances intra-class signal retention while filtering irrelevant noise.

Furthermore, LumiNet's entropy increases to 1.26 (vs. 0.42 for KD), confirming our theoretical claim that normalized logits mitigate overconfidence and retain knowledge about non-target classes. This supports our assertion that batch normalization amplifies class-relevant signals while suppressing extraneous variance, effectively preserving dark knowledge for improved distillation.

**3. Convergence Rate:** Figure 4(c) illustrates LumiNet's faster convergence, achieving a 75% loss reduction within 50 epochs, compared to KD's 60%. This aligns with Theorem 2's linear convergence guarantee under the Polyak-Lojasiewicz (PL) condition, enabled by LumiNet's stabilized gradients.

Additionally, curvature analysis of the Hessian spectrum estimates that LumiNet's PL constant is $2.1\times$ larger than KD's, implying accelerated optimization. This result is particularly significant for resource-constrained deployments, where faster convergence translates to reduced computational cost and improved training efficiency.

## A.7 On the Invariance of Perception under Matching Batch Sizes

**Theorem 3** (Batch Size Dependecy)**.** *Assume the teacher $f^T$ and student $f^S$ each form "perception" logits via class-wise mean and variance computed on the* same *batch size $m$. Then, for each batch $\mathcal{B}$, the KL*

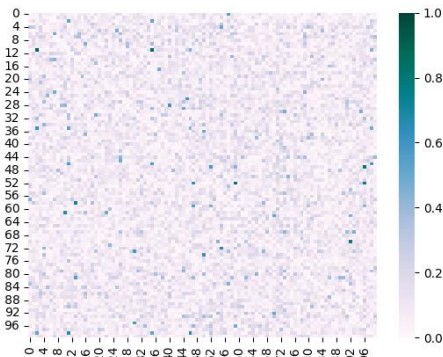
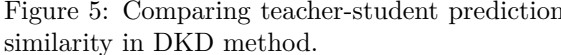

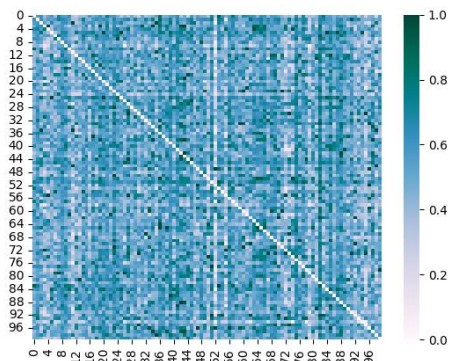

Figure 5: Comparing teacher-student prediction similarity in DKD method.

Figure 6: Comparing Teacher-Student Predictions Similarity in Our Method

*divergence between their perception distributions,*

$$\sum_{x_i \in \mathcal{B}} \text{KL}\left(\sigma\left(\tfrac{h_i^T}{\tau}\right) \,\Big\|\, \sigma\left(\tfrac{h_i^S}{\tau}\right)\right),$$

*cannot systematically degrade when $m$ changes, aside from $\mathcal{O}\left(\tfrac{1}{\sqrt{m}}\right)$ sampling fluctuations. In particular, using the same $m$ for teacher and student preserves the invariance of their class-wise normalization scales and does not hamper distillation performance.*

*Sketch of Proof.* By definition, the teacher's perception logits $h_j^T(x_i) = \left(z_j^T(x_i) - \mu_j^T\right)/\sigma_j^T$ depend on $(\mu_j^T, \sigma_j^T)$, computed over the same $m$ samples as the student's $(\mu_j^S, \sigma_j^S)$. When $m$ changes, both $(\mu_j^T, \sigma_j^T)$ and $(\mu_j^S, \sigma_j^S)$ shift by $\mathcal{O}\left(\tfrac{1}{\sqrt{m}}\right)$ due to sampling variance. Hence the *relative* teacher–student scaling remains consistent.

Formally, let $\Delta_j^T(x_i) = z_j^T(x_i) - \mu_j^T$ and $\Delta_j^S(x_i) = z_j^S(x_i) - \mu_j^S$. Both are divided by $\sigma_j^T, \sigma_j^S$ that are likewise estimated from $m$ samples. Thus, scaling in $h^T$ and $h^S$ aligns in expectation, keeping $\text{KL}\left(\sigma(h^T/\tau),\, \sigma(h^S/\tau)\right)$ invariant up to $\mathcal{O}\left(\tfrac{1}{\sqrt{m}}\right)$. Consequently, matching batch sizes for teacher and student ensures that the perception-based distillation loss does not degrade simply due to changing $m$. $\qquad\square$

### A.8 Mimicking Perception rather than Raw Logits.

The primary goal of traditional knowledge distillation is to replicate the raw logits of the teacher, as illustrated in Figure 5. This figure demonstrates that the predictions closely resemble the teacher's logits. In this method, we often face overconfidence issues, resulting in inferior performance compared to feature-based KD. Moreover, despite our aim to mimic the logits of the teacher, a substantial gap persists between teachers and students. However, in our approach, *LumiNet*, depicted in Figure 6, the prediction similarity to the teacher's logits is significantly lower compared to DKD. Yet, as detailed in the main paper, *LumiNet* achieves better performance scores than DKD (Zhao et al., 2022) also, the gap between teacher and student is minimized. Importantly, teacher and student predictions are independent, diverging from similar logits. This indicates that, in logit-based distillation, we can achieve superior performance without directly mimicking the raw logits. Also, within the parameters, the student models are capable of independently learning features through their innate pattern recognition abilities without being explicitly guided to mimic the pattern learning process of the teacher model. Consequently, this empowers the student model to create new representations and inter-class relationships, for instance, a capability that traditional knowledge distillation methods lack.

### A.9    Implementation Details

For a fair comparison, we maintain a similar setup to previous methods(Hinton et al., 2015; Zhao et al., 2022). In traditional KD(Hinton et al., 2015), both Cross-Entropy loss and Kullback-Leibler (KL) Divergence loss are employed. Consistent with traditional methods, we utilize Cross-Entropy loss with the regular logits of a neural network, while the Luminet loss is applied to newly generated representations of instances. Further details of the Luminet loss are provided in the main paper. In this scenario, the hyperparameter $\alpha$ is set such that $\alpha > t^2$, where $\alpha$ represents a constant associated with the Luminet loss when combined with Cross-Entropy loss and $t$ represents temperature. Specific implementation details for each task are outlined below.

**Image Recognition:** For training a student model on the CIFAR-100 dataset, we use a batch size of 64 and train for a total of 240 epochs. The initial learning rate (LR) is set to 0.05, with learning rate decay applied at epochs 150, 180, and 210, where the LR is reduced by a factor of 0.1 each time. We employ a weight decay of 0.0005 and a momentum of 0.9 in our stochastic gradient descent (SGD) optimizer.

When training on the ImageNet dataset, we use a batch size of 512 and train for a total of 100 epochs. The initial LR is set to 0.2, with learning rate decay scheduled at epochs 30, 60, and 90, where the LR is decreased by a factor of 0.1 each time. We apply a weight decay of 0.0001 and utilize a momentum of 0.9 in the SGD optimizer.

**Object Detection:** For training object detection student models on the MS-COCO 2017 dataset, we use an image per batch of 8. The base learning rate is set to 0.01, and the maximum number of iterations is set to 180,000. Learning rate decay is applied at specific steps during training, with decay steps set at 120,000 and 160,000 iterations.

**Vision Transformer** We adopt the settings described in reference (Li et al., 2022b) for training the student model of vision transformer variants. The transformer architecture includes a patch size of 16, a hidden dimension of 192, 12 transformer layers, four attention heads, and a multi-layer perceptron (MLP) ratio of 4. We set the dropout rate to 0.0, the drop path rate to 0.1, and the attention dropout rate to 0.0. For optimization, we use the AdamW optimizer with a base learning rate of $5.0 \times 10^{-4}$ and a minimum learning rate of $5.0 \times 10^{-6}$. The learning rate policy is cosine annealing (cos) with a maximum of 300 epochs. We apply a weight decay of 0.05, a warm-up factor of 0.001, and warm-up epochs of 20.

**Glue Benchmark:** Our experiments are conducted on the GLUE benchmark using the same overall setup as the distillation framework presented by (Wu et al., 2023). Specifically, we utilize their training pipeline, model configurations, and hyperparameter search space without modification. The teacher model is a fine-tuned BERT$_b ase$, while the student model is a compact 6-layer BERT variant introduced by (Turc et al., 2019), matching the architecture used in their work. For all distillation runs, we retain the original search space for learning rate, temperature, and distillation weights. The only change introduced in our setup is the application of our proposed method on top of the standard knowledge distillation process. We replace the raw teacher logits with our perception-normalized logits during training.

### A.9.1    Calibration Analysis

To comprehensively evaluate the calibration properties of LumiNet, we compute three key metrics: Expected Calibration Error (ECE), Maximum Calibration Error (MCE), and False Positive Rate at 95% True Positive Rate (FPR95). These metrics assess how well the model's confidence scores align with actual correctness and its ability to distinguish between correct and incorrect predictions.

**Expected Calibration Error (ECE):**  ECE quantifies the overall miscalibration by measuring the discrepancy between model confidence and accuracy across multiple confidence bins. Predictions are grouped into 15 equally spaced bins based on confidence scores. Within each bin, we compute the average confidence and the actual accuracy. The ECE is then calculated as a weighted sum of the absolute differences between accuracy and confidence, where the weight corresponds to the proportion of samples in that bin. A lower ECE value indicates that the model's predicted probabilities better reflect the true likelihood of correctness.

Table 11: Performance of our method with the incorporation of ReviewKD Loss on CIFAR-100 dataset

| Teacher/Student Architecture | ReviewKD | Ours | Ours* |
|---|---|---|---|
| WRN-40-2 → ShuffleNet-V1 | 77.14 | 76.95 | **77.29** |
| ResNet32×4 → ShuffleNet-V2 | 77.78 | 77.55 | **77.93** |

Table 12: Detection results on MS-COCO using Faster-RCNN-FPN (Lin et al., 2017) backbone with incorporating ReviewKD.

| | | | Feature-Based Methods | | | Logit-Based Methods | | | | |
|---|---|---|---|---|---|---|---|---|---|---|
| | | | \multicolumn{3}{c|}{} | | | | | | |
| \multicolumn{11}{c}{**ResNet101 (Teacher) and ResNet18 (Student)**} | | | | | | | | | | |
| | Teacher | Student | FitNet | FGFI | ReviewKD | KD | TAKD | DKD | Ours | Ours* |
| AP | 42.04 | 33.26 | 34.13 | 35.44 | 36.75 | 33.97 | 34.59 | 35.05 | 35.34 | **36.89** |
| $AP_{50}$ | 62.48 | 53.61 | 54.16 | 55.51 | 56.72 | 54.66 | 55.35 | 56.60 | 56.82 | **57.05** |
| $AP_{75}$ | 45.88 | 35.26 | 36.71 | 38.17 | 34.00 | 36.62 | 37.12 | 37.54 | 37.56 | **39.59** |
| \multicolumn{11}{c}{**ResNet50 (Teacher) and MobileNet-V2 (Student)**} | | | | | | | | | | |
| AP | 40.22 | 29.47 | 30.20 | 31.16 | 33.71 | 30.13 | 31.26 | 32.34 | 32.38 | **34.18** |
| $AP_{50}$ | 61.02 | 48.87 | 49.80 | 50.68 | 53.15 | 50.28 | 51.03 | 53.77 | 53.84 | **53.95** |
| $AP_{75}$ | 45.88 | 30.90 | 31.69 | 32.92 | 36.13 | 31.35 | 33.46 | 34.01 | 33.57 | **36.44** |

**Maximum Calibration Error (MCE):** MCE identifies the worst-case miscalibration by determining the maximum absolute difference between accuracy and confidence across all bins. Unlike ECE, which provides a weighted average measure, MCE focuses on the most severe miscalibration present in any confidence range. This metric is particularly useful for identifying whether the model is drastically over- or under-confident in specific confidence intervals.

**False Positive Rate at 95% True Positive Rate (FPR95):** FPR95 is a robustness metric that evaluates the model's ability to distinguish between true and false positives. It measures the false positive rate when the true positive rate (TPR) is fixed at 95%. The calculation is performed using the Receiver Operating Characteristic (ROC) curve, where the threshold is adjusted such that the TPR reaches 95%, and the corresponding false positive rate is recorded. Lower FPR95 values indicate improved robustness, as fewer incorrect samples are classified with high confidence.

**Implementation Details:** We implement these calibration metrics following standard evaluation procedures:

- ECE and MCE Calculation: We use 15-bin equal-width binning. Predictions are grouped into bins based on confidence scores, and we compute accuracy and confidence within each bin. The ECE and MCE are derived from these statistics. - FPR95 Calculation: We apply the ROC curve method to compute the false positive rate at a fixed true positive rate of 95%. This involves converting softmax probabilities into binary classification labels per class and analyzing class-wise ROC curves.

### A.10 Incorporating with feature-based distillation

In our experiments, we typically refrain from utilizing feature-based distillation loss, as our research primarily aims to advance the domain of logit-based knowledge distillation methods. However, in certain architectures, and to explore its compatibility with existing feature-based KD methods, we incorporated the feature-based loss (ReviewKD (Chen et al., 2021b)) alongside our *LumiNet* loss.

This combination resulted in significant performance improvements, as demonstrated in Table 11 for the image recognition task and Table 12 for the object detection task. In the tables, the asterisk (*) denotes the utilization of the combined loss function. Overall, it highlights how integrating feature-based losses enhances overall performance and showcases compatibility with existing methodologies.

Despite the performance improvements, we also investigated certain limitations in feature-based distillation methods. These methods often require longer convergence times, which deterred us from incorporating

feature-based KD. For instance, ReviewKD (Chen et al., 2021b), despite its comprehensive approach, requires significant training time due to its multi-level distillation process and complex components like the Attention-Based Fusion module. OFD (Cho & Hariharan, 2019), while focusing on multi-layer distillation, demands extra convolutions for feature alignment, increasing computational needs. Similarly, CRD (Tian et al., 2020) employs a contrastive loss that requires a large memory bank, adding to computational costs.

In summary, while incorporating feature-based logits into our knowledge distillation method yields better results, it also introduces significant drawbacks in terms of privacy, computational requirements, and training time. Hence, we advocate for logit-based knowledge distillation as a more resource-efficient and versatile alternative for various applications.

Table 13: Comparison of top-1 and top-5 accuracy and ECE of knowledge distillation and our method under noisy settings across different teacher-student architectures.

| Teacher (Acc.) | Student (Acc.) | Method | Top-1 | Top-5 | ECE |
|---|---|---|---|---|---|
| ResNet32x4 (55.57) | ResNet8x4 (52.21) | KD | 53.88 | **78.95** | 0.16 |
| | | Ours | **54.70** | **79.18** | **0.12** |
| VGG13 (53.77) | VGG8 (51.75) | KD | 54.78 | 78.05 | 0.27 |
| | | Ours | **55.06** | **79.96** | **0.09** |

## A.11 Effect of Overconfidence

To evaluate performance under realistic label noise, we constructed a noisy CIFAR-100 dataset using instance-dependent label noise rather than random corruption Ju et al. (2022). Specifically, we trained a ResNet-18 for two epochs on the clean training set and used its cross-entropy loss to identify the 30% most misclassified training examples. We then replaced their ground-truth labels with the weak model's predictions, simulating plausible human annotation errors. Under this challenging noisy setting, our method significantly outperforms traditional KD across two teacher-student architecture pairs. For the ResNet32x4–ResNet8x4 pair, our approach improves Top-1 accuracy and reduces ECE from 0.16 to 0.12, demonstrating both better accuracy and confidence calibration. Similarly, for VGG13–VGG8, we achieve a 1.5% gain in Top-1 accuracy and drastically reduce ECE from 0.27 to 0.09 (shown in the table 13 ). These results show that our method is more robust under noisy supervision and mitigates the overconfidence often seen in traditional KD approaches.

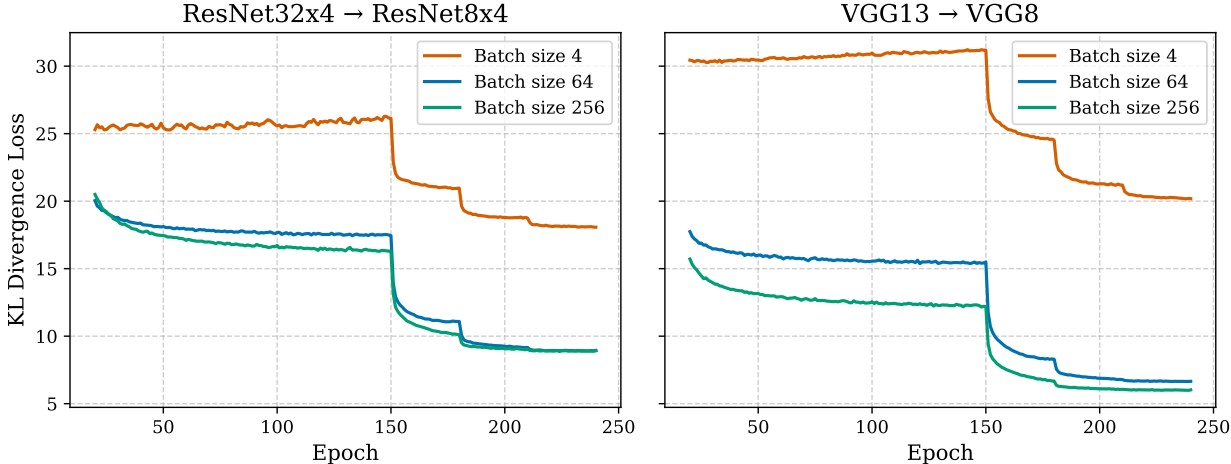

Figure 7: KL divergence loss between teacher and student logits during training for varying batch sizes (4, 64, 256) across two teacher-student pairs: ResNet32x4 → ResNet8x4 and VGG13 → VGG8. All losses are scaled equally for visual comparability.

### A.12 Effect of Batch Size

As shown in Figure 7, we examine the effect of batch size on the KL divergence loss between teacher and student logits during training. For batch sizes of 64 and 256, the KL divergence curves remain stable and nearly identical, indicating that even moderate batch sizes provide sufficient sample diversity to generate reliable batch-level statistical perception, resulting in consistent representations between teacher and student. However, when the batch size is reduced to 4, we observe a noticeable increase and volatility in KL divergence. This is due to the limited number of samples per batch, which weakens the quality of class-level statistical signals and leads to a greater mismatch in logit distributions. This divergence negatively impacts optimization and highlights a limitation of our method in extremely low batch size settings. All KL divergence values were scaled equally for fair visual comparison.

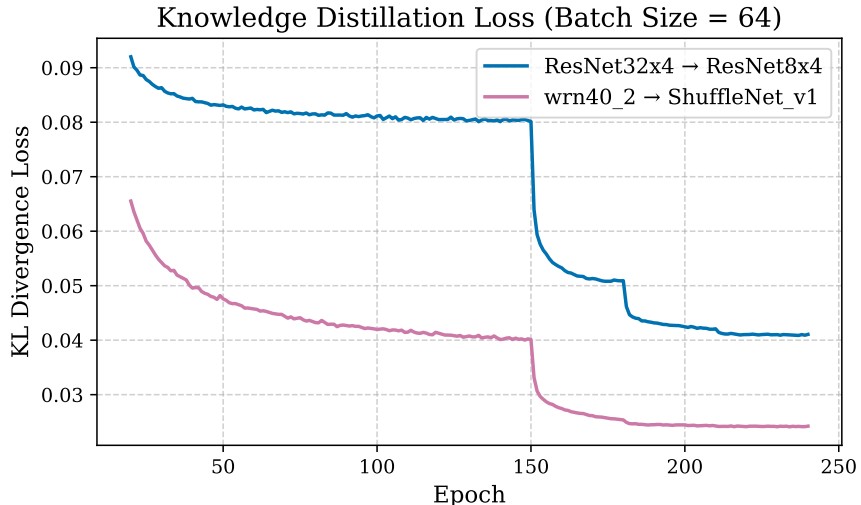

Figure 8: KL divergence loss comparison across different teacher-student configurations with batch size 64. Homogeneous pairs (e.g., ResNet32x4 → ResNet8x4) show lower divergence than heterogeneous ones (e.g., WRN40_2 → ShuffleNet_v1).

Table 14: Top-1 Accuracy (%) on CIFAR-100 for different teacher-student model pairs. 'KD' refers to standard knowledge distillation. 'Ours' uses perception-normalized logits based on local batch statistics, while 'Ours++' replaces local statistics with global teacher statistics aggregated across the training set.

| Teacher → Student | Teacher | Student | KD | Ours | Ours++ |
|---|---|---|---|---|---|
| ResNet32x4 → ResNet8x4 | 79.42 | 72.50 | 73.33 | **77.50** | 76.82 |
| VGG13 → VGG8 | 74.64 | 70.36 | 73.98 | 74.94 | **75.11** |
| WRN-40-2 → ShuffleNetV1 | 75.61 | 70.50 | 74.83 | **76.95** | 76.62 |

### A.13 Global vs Local statistics

To investigate the impact of normalization scope in our distillation method, we compare two variants: (1) one using batch-wise (local) statistics (Ours) and another (2) using dataset-wide (global) statistics (Ours++) for normalizing teacher logits. For the global variant, we precompute the mean and variance of teacher logits over the entire CIFAR-100 training set and apply them consistently during training. In contrast, the local variant computes these statistics dynamically within each batch. As shown in Table 14, both variants outperform the standard KD baseline across all teacher-student pairs, demonstrating the effectiveness of our logit normalization approach. Notably, the local version achieves the best performance, suggesting that adapting to the distribution of each batch introduces beneficial regularization. The global version

remains competitive, indicating that both strategies enhance distillation, though local statistics offer a slight advantage in accuracy.

## A.14 Logit-Space Misalignment

In knowledge distillation settings where the teacher and student networks differ architecturally, we observe that KL divergence loss tends to remain higher throughout training compared to homogeneous configurations, as shown in Figure 8. This is because in heterogeneous setups, the representation of a sample in the logit space is typically not aligned between the teacher and student; architectural differences induce mismatched inductive biases, feature extraction hierarchies, and semantic decompositions. As a result, the student struggles to mimic the teacher's output distribution effectively, even under the same temperature and training hyperparameters. This structural misalignment limits the efficacy of logit-based distillation alone, which relies on the student approximating the teacher's soft targets directly. In contrast, feature-based distillation approaches can partially recover this gap by aligning intermediate feature representations, which are often more robust to architectural divergence and carry richer localized information. This explains why heterogeneous distillation with only logits underperforms, despite sharing the same optimization regime.

## A.15 Logit Complexity Analysis

Neural knowledge distillation faces inherent challenges due to the architectural capacity gap between teacher and student models, where students with fewer parameters struggle to directly mimic the complex distributions generated by larger teachers. Two critical issues arise in traditional KD. First, there is a significant disparity between the probabilities of target and non-target classes. The teacher model tends to produce overly confident predictions for the target classes, which creates a considerable learning burden for the student model, as discussed in section 3.1 of the paper. Second, this challenge intensifies with an increasing number of classes, manifesting as multiple high-probability regions (multi-mode) across the class space. These issues become particularly pronounced in large language models, where the vocabulary size far exceeds typical image classification tasks, resulting in substantially more complex probability distributions for the student to learn. Our perception-based approach effectively addresses these limitations by significantly reducing the class dwarfing effect and diminishing the multi-mode peaks, as demonstrated in Figure 9. Using ResNet18 on ImageNet (1000 classes), we observe that our method produces more balanced probability distributions compared to temperature-scaled KD (T=4), making the dark knowledge transfer more tractable for the student model while preserving essential class relationships.

## A.16 LumiNet in Large-Language Model

KD in Large Language Models (LLMs) presents unique challenges compared to its application in computer vision tasks. In vision models, the logit distribution usually displays a single-mode pattern, making it relatively easy for student models to replicate the teacher's probability distribution. However, LLMs operate with vocabulary spaces that span thousands to millions of tokens, resulting in complex 'multi-mode' distributions for a sample. This fundamental difference makes traditional KD approaches less effective for LLMs.

We used the dataset split within this space for our experiment [2]. We have used 13.5k samples from the Dolly dataset for fine-tuning, while 500 samples were reserved for testing. Additionally, 80 and 240 samples were used from Vicuna and SelfInst, respectively for evaluation. We have adapted our method to make it suitable for LLMs. Our experimental results, as shown in Table 15, demonstrate that our method consistently outperforms existing KD approaches across different model sizes. For instance, with GPT-2 340M as the student model, our method achieves 27.8, 13.8, and 17.1 R-L scores on Dolly, SelfInst, and Vicuna test sets, respectively, surpassing both conventional KD (25.0, 12.0, 15.4) and Sequential KD. Notably, in several cases, our student models even outperform the 1.5B teacher model.

For our experimental setup, we used a 1.5B parameter model as the teacher and tested student models of varying sizes (120M, 340M, and 760M parameters) based on the GPT-2 architecture. The training

---

[2]https://huggingface.co/MiniLLM

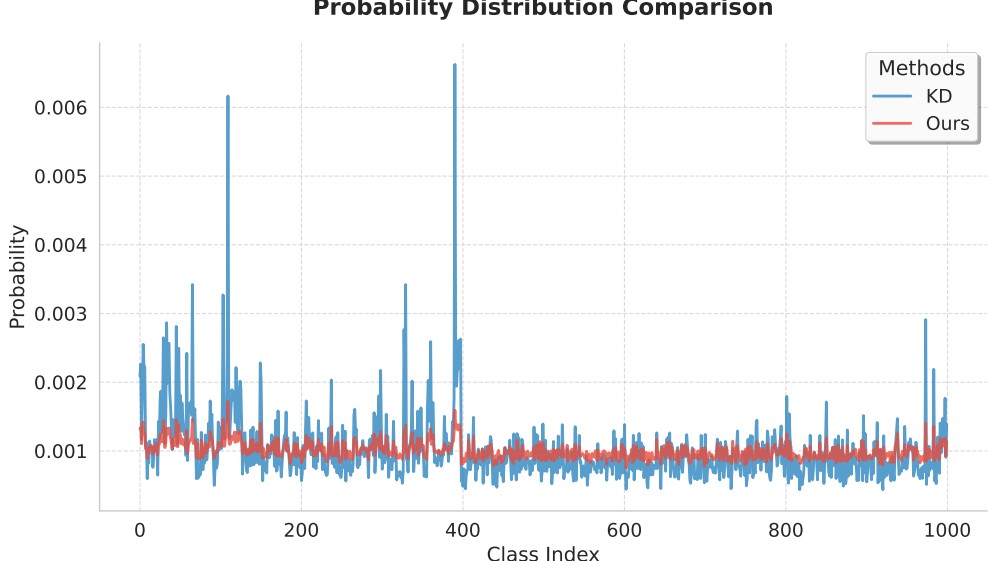

Figure 9: Comparison of probability distributions between traditional KD and our proposed method on ImageNet using ResNet18. The blue line represents temperature-scaled KD (T=4), showing multiple high-confidence regions and significant disparities between target and non-target classes. The red line shows our method's distribution, which effectively reduces both the class dwarfing effect (lower peaks) and multi-modal nature of the distribution, resulting in more manageable class relationships for the student model to learn. This visualization demonstrates how our approach simplifies the dark knowledge transfer while maintaining informative class relationships across the 1000 ImageNet classes.

Table 15: Evaluation results. We report the average R-L scores across 5 random seeds. The best scores of each model size are boldfaced, and the scores where the student model outperforms the teacher are marked with *.

| Model | #Params | Method | Dolly | SelfInst | Vicuna |
|---|---|---|---|---|---|
| Teacher | 1.5B | - | 27.6 | 14.3 | 16.3 |
| GPT-2 | 120M | SFT w/o KD | 23.3 | 10.0 | 14.7 |
| | | KD | 22.8 | 10.8 | 13.4 |
| | | SeqKD | 22.7 | 10.1 | 14.3 |
| | | **Ours** | **23.8**$_{(0.37)}$ | **11.4**$_{(0.42)}$ | **14.9**$_{(0.10)}$ |
| | 340M | SFT w/o KD | 25.5 | 13.0 | 16.0 |
| | | KD | 25.0 | 12.0 | 15.4 |
| | | SeqKD | 25.3 | 12.6 | 16.9* |
| | | **Ours** | **27.8***$_{(0.47)}$ | **13.8**$_{(0.48)}$ | **17.1***$_{(0.16)}$ |
| | 760M | SFT w/o KD | 25.4 | 12.4 | 16.1 |
| | | KD | 25.9 | 13.4 | 16.9* |
| | | SeqKD | 25.6 | 14.0 | 15.9 |
| | | **Ours** | **28.6***$_{(0.49)}$ | **14.7***$_{(0.19)}$ | **17.5***$_{(0.10)}$ |

was conducted with a batch size of 2. We implemented sequence-level tokenization and used the AdamW optimizer with a learning rate of 5e-5. The training was performed on a single 4090 GPU.

## A.17 Broader Impact

LumiNet enhances logit-based knowledge distillation by avoiding the use of intermediate feature representations, which improves privacy and efficiency—especially in scenarios involving commercial APIs or federated

learning. However, caution is warranted in high-stakes applications (e.g., healthcare, surveillance), as output logits can still leak sensitive training information. We strongly encourage responsible use, including thorough calibration, bias analysis, and evaluation of potential misuse in downstream contexts. Future iterations could integrate built-in safeguards or metrics to assess and mitigate such risks.

