# OpenReview forum: "LumiNet: Perception-Driven Knowledge Distillation via Statistical Logit Calibration"
_TMLR — Accepted by TMLR_

### Review · Reviewer_F6Zz · 2025-05-09

**Summary Of Contributions:**

This paper explores the logit-based knowledge distillation from the perspective of statistical logit calibration. Authors argue that logit-based distillation have the challenge of model's overconfidence and confirmation bias, which leads to inferior performance
compared to feature-based methods. To bridge this gap, authors present LumiNet, a novel knowledge distillation algorithm designed to enhance logit-based distillation. By normalizing the batches of samples, the propose LumiNet can improve the distillation performance. Extensive experiments shows LumiNet's effectiveness and mitigate the corresponding challenges.

**Audience:**

Yes

**Broader Impact Concerns:**

None.

**Claims And Evidence:**

Yes

**Requested Changes:**

See Weaknesses.

**Strengths And Weaknesses:**

Strengths:
1. Exploring logit-based knowledge distillation helps the deployment of small models.

2. The proposed method is simple and effective.

Weaknesses:
1. Differences between overconfidence and confirmation bias are not clear. As described in the main body, confirmation bias is based on overconfidence and both harms the distillation performance.

2. Why does the propose method being called LumiNet? This is not an abbreviation of some phrases, nor is it a network structure.

3. Lack of comparison of knowledge distillation methods on 2025.

4. The taxonomy of knowledge distillation needs discuss relation-based ditsillation. RKD and this paper belongs to this category.

---

> ### Author Response · Authors · 2025-06-09
>
> We sincerely thank the reviewer for the positive feedback on our methods.
> We have addressed the weaknesses the reviewer raised as follows:
>
> **W1:** *"Differences between overconfidence and confirmation bias are not clear. As described in the main body, confirmation bias is based on overconfidence and both harms the distillation performance."*
>
> **Response:** Thank you for the helpful observation. We have now clarified the distinction between overconfidence and confirmation bias in the revised manuscript (Introduction, pages 1–2). Specifically, we define overconfidence as the teacher model’s tendency to assign excessively high probabilities to its predictions, which may suppress informative uncertainty and destabilize student learning. In contrast, confirmation bias refers to the student’s inclination to mimic the teacher’s predictions—including incorrect or overconfident ones—without sufficient correction, thereby reinforcing errors during distillation [1].
>
> **W2:** *"Why does the propose method being called LumiNet? This is not an abbreviation of some phrases, nor is it a network structure."*
>
> **Response:** Thank you for pointing this out. We have now clarified the naming rationale in Section 3.2. The name LumiNet is inspired by the method’s core intuition—illuminating the "dark knowledge" embedded in the teacher’s logits. Specifically, our approach sheds light on subtle inter-sample relationships by leveraging batch-level statistical perception to calibrate overconfident predictions. While not an acronym, Lumi metaphorically captures this illumination process,the Net in LumiNet signifies the underlying network of relational connections our method constructs among samples during distillation, highlighting its structure-aware and model-agnostic design.
>
> **W3:** *"Lack of comparison of knowledge distillation methods on 2025."*
>
> **Response:** Thank you for the comment. Many of the knowledge distillation methods introduced in 2025 are either task-specific or rely on architectures or settings that are not directly compatible with our experimental setup. However, we included a representative feature-based method from [2] in Table 1 for comparison. Despite this, our method consistently achieves higher accuracy across most teacher-student pairs.
>
> We also evaluated our method across different settings and datasets. Please refer to our responses to Reviewer zjPH and Reviewer w2Kg for detailed comparisons and results.
>
> **W4:** *"The taxonomy of knowledge distillation needs discuss relation-based ditsillation. RKD and this paper belongs to this category"*
>
> **Response:** You are absoulately right and thanks for the comment. Indeed, both RKD [3] and our proposed LumiNet focus on transferring relational knowledge by modeling relationships between samples. However, we categorize RKD as a feature-based method because it relies on pooled embeddings from the teacher and student networks, aligning pairwise distances and triplet-wise angles in the feature space using smooth L1 loss. In contrast, LumiNet operates directly in the logit space by modeling relational structures through batch-level statistical perception. We have added this clarification in the 'comparison methods' (section 4.2) on page 9.
>
> **References**
> 1. Zhang, Weijia, et al. "Cross-View Consistency Regularisation for Knowledge Distillation." Proceedings of the 32nd ACM International Conference on Multimedia. 2024.
> 2. Lu, Guoming, et al. "BDCKD: Unlocking the Power of Brownian Distance Covariance in Knowledge Distillation." ICASSP 2025-2025 IEEE International Conference on Acoustics, Speech and Signal Processing (ICASSP). IEEE, 2025.
> 3. Park, Wonpyo, et al. "Relational knowledge distillation." Proceedings of the IEEE/CVF conference on computer vision and pattern recognition. 2019.

---

### Review · Reviewer_w2Kg · 2025-05-12

**Summary Of Contributions:**

This paper proposes **LumiNet**, a new approach to knowledge distillation that revisits logit-based methods with a twist. Unlike prior logit-focused strategies that treat each instance independently, LumiNet introduces the idea of “**perception**”—a form of batch-aware logit calibration. By leveraging batch-level mean and variance statistics per class, the method constructs “perception logits” that temper overconfidence and reduce confirmation bias. This is inspired by Gestalt psychology, particularly Kurt Lewin’s field theory, where each instance is influenced by the context of surrounding samples. LumiNet is shown to work well across several tasks—image classification, object detection, and transfer learning—outperforming both standard and state-of-the-art feature-based KD techniques, without incurring additional computational costs or architectural complexity.

**Audience:**

Yes

**Broader Impact Concerns:**

The paper does not raise significant ethical red flags. In fact, by avoiding the use of intermediate representations, LumiNet offers greater privacy protection in scenarios where teacher internals are sensitive or inaccessible—such as in federated learning or model distillation from commercial APIs.

However, some implicit risks are worth noting:

- Logits can still carry latent information about training data, especially in high-stakes domains (e.g., healthcare, surveillance), which may warrant caution.

- The improved efficiency might encourage deployment in low-resource or under-regulated environments, where calibration and fairness checks are often overlooked.

If not already included, a brief Broader Impact Statement addressing these points—especially around calibration, privacy, and misuse—would be a helpful addition.

**Claims And Evidence:**

Yes

**Requested Changes:**

**Essential:**

- Clarify limitations of LumiNet more explicitly. For example, what happens when batch sizes are small, or classes are highly imbalanced?

- Include a more formal justification (e.g., statistical or optimization perspective) for why batch-wise normalization of logits benefits the KD process beyond mitigating overconfidence.

**Optional but Recommended:**

- Include a simple NLP experiment—even if limited—to support generalization claims.

- Provide more analysis on failure cases or scenarios where LumiNet underperforms compared to feature-based methods.

- Consider evaluating the method in semi-supervised or noisy label settings, where overconfidence might be even more detrimental.

**Strengths And Weaknesses:**

**Strengths:**
- The core idea of perception logits is original and intuitive, presenting a practical means to contextualize logits using batch-level information.

- It delivers consistent empirical gains across various datasets (CIFAR-100, ImageNet, MS-COCO) and architectures (ResNets, MobileNet, ViTs), showing both robustness and scalability.

- Unlike many recent distillation methods, it introduces no additional trainable parameters and keeps latency on par with standard KD.

- The authors provide solid ablations and calibration analyses, demonstrating reduced sensitivity to temperature and improved resistance to confirmation bias.

- The framework is architecture-agnostic, successfully extending to vision transformers, a domain where feature-based methods often dominate.

**Weaknesses:**

- The theoretical underpinnings—particularly around how batch normalization of logits improves generalization—are more heuristic than formal. The Gestalt theory analogy, while creative, lacks rigorous justification.

- All experiments are within computer vision. Despite claims of broader relevance, no non-visual domain results (e.g., in NLP) are shown.

- The method relies on sufficiently large and diverse batches to function optimally. Its performance in low-data or imbalanced class settings remains unclear.

- Some results, though strong, could benefit from direct statistical significance tests to distinguish gains from noise across trials.

---

> ### Author Response · Authors · 2025-06-09
> **Requested Changes (RC) Essential**
>
> Thank you very much for your thoughtful and encouraging comments. We appreciate your recognition of the originality and practical impact of our LumiNet approach.
>
> We have addressed the changes the reviewer requested accordingly, and our detailed responses are provided below.
>
> **Requested Changes (RC) Essential:**
>
> **RC1:** *"Clarify limitations of LumiNet more explicitly. For example, what happens when batch sizes are small, or classes are highly imbalanced?"*
>
> **Response:**  Thank you for the helpful suggestion. While we previously included a batch size comparison in our ablation study, we have now conducted a more detailed analysis as per your request. Our findings indicate that when the batch size is very small (e.g., 4) on datasets with a large number of classes, such as CIFAR-100, the batch-level statistical perception becomes unreliable due to insufficient class diversity. This results in unstable and noisy representations in the logit space, leading to a distributional mismatch between teacher and student logits. As shown by the increased KL divergence loss in Figure 7 ( Section A.12), this mismatch adversely affects optimization and distillation performance. We acknowledge this limitation and view it as a valuable direction for future enhancement.
>
> **RC2:** *"Include a more formal justification (e.g., statistical or optimization perspective) for why batch-wise normalization of logits benefits the KD process beyond mitigating overconfidence."*
>
> **Response:**  Thank you for the suggestion. While a theoretical analysis was already provided in the appendix, we have now added a formal justification in Section 3.2, as per your suggestion. This theoretical justification shows that our method performs a variance-stabilizing transformation on logits, which acts as a form of gradient preconditioning. Specifically, it improves optimization conditioning and ensures uniform gradient variance across classes. We also found that this aligns with prior theoretical work—such as Menon et al. [1] on variance reduction in soft-label distillation—and connects to the Neural Collapse phenomenon [2], further supporting the robustness and generality of our approach.

---

> ### Author Response · Authors · 2025-06-09
> **Requested Changes (RC) Optional or Reccomended**
>
> **RC3:** *"Include a simple NLP experiment—even if limited—to support generalization claims."*
> **Response:**  As per your recommendation, we conducted a preliminary experiment on the GLUE benchmark within the limited available time to test the generalization of our method beyond vision tasks. Interestingly, our method outperforms competing KD techniques on 7 out of 8 GLUE tasks using a BERT<sub>6 </sub>  student model. This result supports the robustness and general applicability of LumiNet in the NLP domain. The corresponding results have been added to the main paper (page 10, table 4) and are shown in the table below. Furthermore, we had previously evaluated our method on an instruction-following dataset using a small language model (GPT-2), as shown in Table 15 (Appendix A.16), further supporting the cross-domain effectiveness of our approach.
>
> Table 1: Performance comparison of student models distilled from BERT₍bₐₛₑ₎ on the GLUE benchmark.
> | Model                       | #Params | CoLA (Mcc) | MNLI-m/mm (Acc) | SST-2 (Acc) | QNLI (Acc) | MRPC (F1) | QQP (Acc) | RTE (Acc) | STS-B (Spear) | Avg  |
> |----------------------------|---------|------------|------------------|-------------|-------------|------------|-------------|------------|----------------|-------|
> | BERT₍bₐₛₑ₎  (Teacher) | 110M   | 60.3       | 84.9 / 84.8       | 93.7        | 91.7        | 91.4       | 91.5        | 69.7       | 89.4           | 84.1  |
> | BERT-6 (Student)    | 66M    | 51.2       | 81.7 / 82.6       | 91.0        | 89.3        | 89.2       | 90.4        | 66.1       | 88.3           | 80.9  |
> | KD                         | 66M    | 53.6       | 82.7 / 83.1       | 91.1        | 90.1        | 89.4       | 90.5        | 66.8       | 88.7           | 81.6  |
> | PD                         | 66M    | --         | 82.5 / 83.4       | 91.1        | 89.4        | 89.4       | 90.7        | 66.7       | --             | --    |
> | PKD                        | 66M    | 45.5       | 81.3 / --         | 91.3        | 88.4        | 85.7       | 88.4        | 66.5       | 86.2           | 79.2  |
> | TinyBERT                   | 66M    | 53.8       | 83.1 / 83.4       | 92.3        | 89.9        | 88.8       | 90.5        | 66.9       | 88.3           | 81.7  |
> | CKD                        | 66M    | 55.1       | 83.6 / 84.1       | **93.0**        | 90.5        | 89.6       | 91.2        | 67.3       | 89.0           | 82.4  |
> | MGSKD                      | 66M    | 49.1       | 83.3 / 83.9       | 91.7        | 90.3        | 89.8       | 91.2        | 67.9       | 88.5           | 81.5  |
> | **Ours**                   | 66M    | **55.8**   | **83.72 / 84.23** | 91.3        | **90.7**    | **89.9**   | **91.6**    | **69.7**   | **89.3**       | **83.0** |
> | *Δ (vs. best)*             | —      | +2.2       | +1.02 / +1.13     | +0.2        | +0.6        | +0.5       | +1.1        | +2.9       | +0.6           | +1.4  |
>
> **RC4:** *"Provide more analysis on failure cases or scenarios where LumiNet underperforms compared to feature-based methods."*
> **Response:** As shown in Table 1 and Appendix A.14 (Figure 8), our method performs slightly worse in certain heterogeneous teacher-student settings. We suspect this is due to architectural misalignment in logit space, where the student's representation struggles to accurately mimic the teacher's output distribution. Unlike feature-based methods that leverage intermediate layers to bridge this gap, LumiNet relies solely on final logits, which may be less expressive across diverse architectures. We acknowledge this limitation and view it as a valuable direction for future enhancement.
>
> **RC5:** *"Consider evaluating the method in semi-supervised or noisy label settings, where overconfidence might be even more detrimental."*
> **Response:** Thank you for the suggestion. In Appendix A.11 Table 13, we evaluate our method under instance-dependent label noise settings [3]. We observe that our approach consistently improves confidence calibration (lower ECE) and shows better Top-1 accuracy compared to KD. While performance may vary slightly across architectures, these results suggest our method is more robust to overconfidence in noisy settings—precisely where traditional KD often struggles. These results support our central claim. We appreciate the reviewer for highlighting this important use case that reinforces the strength of our approach.
>
> Table 2: Evaluation under instance-dependent label noise.
> | Teacher (Acc.)        | Student (Acc.)       | Method | Top-1 | Top-5 | ECE  |
> |-----------------------|----------------------|--------|-------|-------|------|
> | ResNet32x4 (55.57)    | ResNet8x4 (52.21)    | KD     | 53.88 | 78.95 | 0.16 |
> |                       |                      | Ours   | **54.70** | **79.18** | **0.12** |
> | VGG13 (53.77)         | VGG8 (51.75)         | KD     | 54.78 | 78.05 | 0.27 |
> |                       |                      | Ours   | **55.06** | **79.96** | **0.09** |

---

> ### Author Response · Authors · 2025-06-09
> **Broader Impact Statement and References**
>
> **Broader Impact Concern:**
> In response to your suggestion, we have included a broader impact statement at the end of the paper (A.17). The placement of this section will be adjusted in the camera-ready version based on the Action Editor’s guidance.
>
> **References**
> 1. Menon, Aditya K., et al. "A statistical perspective on distillation." International Conference on Machine Learning. PMLR, 2021.
>
> 2. Papyan, Vardan, X. Y. Han, and David L. Donoho. "Prevalence of neural collapse during the terminal phase of deep learning training." Proceedings of the National Academy of Sciences 117.40 (2020): 24652-24663.
>
> 3. Ju, Lie, et al. "Improving medical images classification with label noise using dual-uncertainty estimation." IEEE transactions on medical imaging 41.6 (2022): 1533-1546.

---

### Review · Reviewer_zjPH · 2025-05-30

**Summary Of Contributions:**

The paper introduces **LumiNet**, a novel logit-based knowledge distillation method that addresses the limitations of conventional logit matching—namely overconfidence and confirmation bias—by introducing a *perception-driven calibration mechanism*. Instead of treating logits independently, LumiNet normalizes them using batch-level class-wise statistics (mean and variance), forming *perception logits* that reflect contextual relationships across samples. This calibration reduces extreme predictions and better captures inter-class dependencies without requiring intermediate features, preserving privacy and efficiency. Empirical results on CIFAR-100, ImageNet, MS COCO, and TinyImageNet demonstrate that LumiNet consistently outperforms state-of-the-art logit- and feature-based distillation methods across classification, detection, and transfer learning tasks, all while maintaining low computational overhead.

**Audience:**

Yes

**Broader Impact Concerns:**

LumiNet may still inherit and subtly reinforce biases from the teacher model, especially when batch-level calibration is applied without safeguards, potentially affecting fairness in real-world applications.

**Claims And Evidence:**

Yes

**Requested Changes:**

Add more calibration methods for comparison, since your calibration techniques works, how about others?

How about your method applied on standard calibration task, how does it compare to others?

**Strengths And Weaknesses:**

**Strength**
1. Reduces overconfidence and confirmation bias using batch-level logit normalization.
2. Adds no extra parameters and matches vanilla KD in speed.
3. Outperforms state-of-the-art on classification, detection, and transfer tasks across models.

**Weakness**
1. Since calibration can improve KD on logits, how about other calibration methods? Even some models that originally trained by a calibration loss such as focal loss[1,2]?
2. [3] is a recently proposed method that can be applied on top of logit, how's the performance if applied with such methods?
3. Why batch level is better? how about the whole dataset statistic for calibration?
4. Why normallzing can suppressing extreme confidence values ? This can also make under confidence, right?


[1] Calibrating Deep Neural Networks using Focal Loss
[2] Dual focal loss for calibration
[3] Feature Clipping for Uncertainty Calibration

---

> ### Author Response · Authors · 2025-06-09
> **Addressing Requested Changes**
>
> We sincerely thank the reviewer for his/her thoughtful and constructive feedback. We particularly appreciate the emphasis on calibration, which directly aligns with one of our core claims—that effective knowledge distillation benefits from proper calibration. Following your suggestions, we conducted additional experiments incorporating alternative calibration techniques. The results, now included in the revised manuscript and Appendix, further support our claim. Detailed responses to each concern are provided below.
>
> **RC1:** *"Consider evaluating the method in semi-supervised or noisy label settings, where overconfidence might be even more detrimental."*
>
> **Response:** Thank you for the insightful suggestion. As shown in Table 5 ( described in 'Effect of Calibration'), we have added comparisons with three calibration methods: Focal Loss (FL), Dual Focal Loss (DFL), and FCUC. All three approaches improve performance over vanilla KD, reinforcing the growing consensus that calibration is essential for effective distillation. FL and DFL act as alternatives to the cross-entropy loss during training, while FCUC operates directly on the logits post-training. Interestingly, even simple loss replacements like FL and DFL yield better accuracy and calibration, suggesting that calibration-aware objectives are a promising research direction. We also combined our method with DFL and observed further gains, though our standalone method with CE still performs best overall. These findings highlight that all three methods—whether loss-based (FL/DFL) or logit-based (FCUC)—are valuable for improving KD and merit further investigation in future work.
>
> Table 1: Accuracy and ECE on CIFAR-100 for different teacher-student distillation setups and methods.
> | Teacher   | Student | KD (Acc/ECE) | KD+FL | KD+DFL | Kd+FCUC | Ours+DFL | Ours |
> |-----------|---------|--------------|-------|--------|------|----------|-------|
> | RN32x4    | RN8x4   | 73.33 / 0.11 | 74.41 / 0.10 | 74.51 / 0.09 | 76.37 / 0.09 | 76.50 / 0.07 | **77.50 / 0.06** |
> | VGG13     | VGG8    | 72.98 / 0.12 | 73.88 / 0.11 | 73.39 / 0.10 | 74.28 / 0.06 | 74.66 / 0.07 | **74.94 / 0.06** |
> | WRN40-2   | SNV1    | 74.83 / 0.13 | 75.26 / 0.12 | 75.11 / 0.11 | 75.02 / 0.09 | **76.98 / 0.04** | 76.95 / 0.07 |
>
> **RC2:** *"Consider evaluating the method in semi-supervised or noisy label settings, where overconfidence might be even more detrimental."*
>
> While our method improves calibration within the KD pipeline, it is not designed to serve as a general-purpose calibration method outside this context. Standard calibration approaches—such as Temperature Scaling, or FCUC —are explicitly constructed to maintain the decision boundary of the model (i.e., the argmax class remains unchanged) while adjusting confidence scores. This property allows them to be applied post-hoc on trained models and used with losses like CE or DFL.
>
> In contrast, our method performs relational normalization of logits across the batch, which changes their geometric structure. Specifically, the logits produced by our method are not guaranteed to preserve the maximum logit at the target class, which breaks compatibility with standard calibration metrics and post-hoc evaluation frameworks that rely on prediction invariance. So, we use these logits to extract knowledge from the teacher via KL-Divergence loss, which does not require the logits to follow the standard distribution typically expected by cross-entropy. Unlike FCUC—which can be seamlessly integrated with CE or DFL because it preserves logit ordering—our logits are scaled and centered to better mimic teacher-student relationships during KD training (equaiton 3). Consequently, they are optimized to support CE loss within KD, but not suitable for plug-and-play calibration tasks.

---

> ### Author Response · Authors · 2025-06-09
> **Addressing Weaknesses**
>
> **W1 and W2:**
>
> **Response:** Please see the response of R1
>
> **W3**: *"Why batch level is better? how about the whole dataset statistic for calibration?"*
>
> **Response:** We thank the reviewers for raising this point. To address this, we have included a new ablation study in Table 14 comparing local (batch-wise) and global (dataset-wide) normalization of teacher logits. We also added a dedicated subsection titled “Global vs Local Statistics” in the revised manuscript (section A.13). Our findings show that while using global statistics provides stability, our empirical results show that batch-wise (local) normalization consistently yields better performance. This is likely because local statistics capture dynamic variations during training, acting as a form of adaptive regularization. As shown in our results, global statistics still improve over the KD baseline, but local adaptation offers a slight yet consistent advantage in accuracy.
>
> Table 2: Top-1 Accuracy (%) on CIFAR-100 for different teacher-student model pairs. ‘KD‘ refers to standard knowledge distillation. ‘Ours‘ uses perception-normalized logits based on local batch statistics, while ‘Ours++‘ replaces local statistics with global teacher statistics aggregated across the training set.
>
> | Teacher → Student             | Teacher | Student | KD    | Ours (Local) | Ours++ (Global) |
> |------------------------------|---------|---------|-------|--------------|-----------------|
> | ResNet32x4 → ResNet8x4       | 79.42   | 72.50   | 73.33 | **77.50**     | 75.92           |
> | VGG13 → VGG8                 | 74.64   | 70.36   | 73.98 | **74.94**     | 73.43           |
> | WRN-40-2 → ShuffleNetV1      | 75.61   | 70.50   | 74.83 | **76.95**     | 76.62           |
>
> **W4**: *"Why normallzing can suppressing extreme confidence values ? This can also make under confidence, right?"*
>
> Yes, you're absolutely right—normalizing logits can indeed suppress both extreme high-confidence and, potentially, unnecessary low confidence. However, our primary concern is the target class, which neural networks naturally assign very high probability to. This behavior arises because the cross-entropy loss encourages the network to inflate the logits for the correct class to minimize loss—studies have shown that { larger logit magnitudes lead directly to overconfident softmax outputs}[1]
>
> By normalizing logits, we effectively control the influence of these extreme magnitudes across all classes. While this could lead to under-confidence in non-target classes (which are less critical), it prevents runaway confidence in the target class—the most significant signal during training.
>
> Furthermore, Our focus on the target class aligns with Decoupled Knowledge Distillation (DKD)[2], which separates distillation into TCKD (target class) and NCKD (non-target classes). DKD highlights that emphasizing the target class is critical for effective student learning—a principle reinforced by our normalization approach.
>
> In short, while normalization can temper confidence globally, its real value lies in stabilizing the signal for the target class, ensuring training focuses on firmly learning the correct labels rather than inflated, uncontrolled probabilities.
>
> **References**
>
> 1. Wei, Hongxin, et al. "Mitigating neural network overconfidence with logit normalization." International conference on machine learning. PMLR, 2022.
> 2. Zhao, Borui, et al. "Decoupled knowledge distillation." Proceedings of the IEEE/CVF Conference on computer vision and pattern recognition. 2022.

---

### Author Response · Authors · 2025-06-09

We sincerely thank all reviewers for their thorough and constructive feedback, which has significantly improved our manuscript. We have carefully addressed all the concerns raised and substantially revised our paper accordingly. All changes are highlighted in blue text in the revised manuscript.

**Summary of the Revisions**

**We have strengthened the theoretical foundation and conceptual clarity by**:
* Clarifying the distinction between overconfidence and confirmation bias (Introduction, pages 1-2) **[Suggested by Reviewer F6Zz]**
* Providing formal theoretical justification with variance-stabilizing transformation analysis (Section 3.2) **[Suggested by Reviewer w2Kg]**
* Explaining LumiNet naming rationale and core intuition of "illuminating dark knowledge" (Section 3.2) **[Suggested by  Reviewer F6Zz]**
* Refining knowledge distillation taxonomy and method categorization (Section 4.2, page 9) **[Suggested by  Reviewer F6Zz]**

**We have conducted extensive additional experimental validation**:
* Cross-domain evaluation with comprehensive GLUE benchmark results (Table 4, page 10) **[Suggested by Reviewer w2Kg]**
* Calibration method comparison with FL, DFL, and FCUC (Table 5) **[Suggested by  Reviewer zjPH]**
* Noisy label robustness evaluation (Appendix A.11, Table 13) **[Suggested by Reviewer w2Kg]**
* Batch size vs. global statistics analysis (Appendix A.13, Table 14) **[Suggested by  Reviewer zjPH]**
* Small batch size limitation analysis with KL divergence study (Section A.12, Figure 7) **[Suggested by Reviewer w2Kg]**

**We have broadened the scope and robustness analysis by**:
* Including failure case analysis for heterogeneous architectures (Appendix A.14, Figure 8) **[Suggested by Reviewer w2Kg]**
* Evaluating performance under instance-dependent label noise settings **[Suggested by Reviewer w2Kg]**
* Comparing with representative 2025 knowledge distillation methods **[Suggested by  Reviewer F6Zz]**
* Adding broader impact statement (Appendix A.17) **[Suggested by Reviewer w2Kg and zjPH]**

**We have enhanced methodological understanding by**:
* Providing statistical justification for local vs. global normalization choice **[Suggested by  Reviewer zjPH]**
* Explaining normalization effects on confidence suppression and target class stabilization **[Suggested by  Reviewer zjPH]**
* Connecting our approach to Neural Collapse phenomenon and variance reduction theory **[Suggested by Reviewer w2Kg]**

Furthermore, we have improved the paper's clarity and comprehensiveness. These revisions thoroughly address all reviewers' concerns while significantly strengthening the paper's theoretical foundation, empirical validation, and practical impact.

---

### Decision · Action_Editor_qtMD · 2025-07-28

**Recommendation:** Accept as is

**Audience:**

Yes

**Audience Explanation:**

Knowledge distillation remains a core interest area for many in the TMLR community spanning vision and NLP domains.

**Claims And Evidence:**

Yes

**Claims Explanation:**

The authors present LumiNet, a novel logit-based knowledge distillation approach using batch-level statistical calibration to mitigate overconfidence and confirmation bias. They support their claims through extensive experiments across diverse benchmarks (CIFAR-100, ImageNet, MS COCO, TinyImageNet, and even a text benchmark GLUE based on reviewer feedback), comparing against both logit-based and feature-based methods. Ablations and robustness tests (e.g., small batch sizes, noisy labels, heterogeneous architectures) further bolster the soundness of the method. All reviewer concerns have been satisfactorily addressed in the revised manuscript.